# Impact of seasonal variation on the oral and nasopharyngeal microbiome in school-aged children: the school MicroBE initiative

Juan Castro-Severyn,[1,2,3] Nicolás Pacheco,[1] Guillermo Valdivia,[4] Gabriel I. Krüger,[1] Coral Pardo-Esté,[1] Francisco Remonsellez,[2,3] Aldo Gaggero,[5] Gloria Arriagada,[6] Fernando Valiente-Echeverria,[7] Jorge Olivares-Pacheco,[8] Jorge H. Valdes,[4] Claudia P. Saavedra[1]

**ABSTRACT**   The "School MicroBE" initiative explores how the built environment impacts microbial communities in school-aged children by examining temporal dynamics and shifts in nasopharyngeal and oral microbiomes. This longitudinal study involved 119 children aged 4 to 13 at a public school, with nasopharyngeal and oral samples collected in autumn, winter, and spring of 2023. Using 16S rRNA gene sequencing, we assessed microbial composition through alpha and beta diversity analyses, characterized microbial assemblages, and evaluated the relative abundance of key taxa. Significant seasonal variations were observed, with an increase in alpha diversity from autumn to spring in nasopharyngeal samples. Beta diversity analyses did not reveal distinct clustering patterns based on collection months. Hierarchical clustering identified four major microbiome groups with characteristic taxonomic distribution, and co-occurrence network analysis suggested both synergistic and competitive interactions among taxa. Longitudinal transition analysis between microbiome clusters revealed dynamic changes over time, providing a baseline of microbiome states in the tested children. These findings highlight the importance of microbial community shifts in the environment by providing direct measures on microbiome stability and diversity in children, providing insights into how microbial communities respond to environmental fluctuations, including potential pathogen exposures. Understanding these temporal changes will improve the development of targeted public health strategies to assess and manage potential infectious disease outbreaks and the emergence of antimicrobial resistance in school settings.

**IMPORTANCE**   The "School MicroBE" initiative enhances our understanding of pediatric microbiome dynamics by revealing temporal and compositional shifts, thus establishing basal studies on a sentinel school contributing to the understanding of pediatric microbiome and its associated health issues.

**KEYWORDS**   oral microbiome, nasopharyngeal microbiome, transition, schoolers, public health

In an era marked by the emergence of new pathogens and the global crisis of antimicrobial resistance, understanding the dynamics of the pediatric microbiome has become increasingly important, especially in built environments. Schools represent a unique and dynamic setting for studying human microbiomes, particularly in children who are in a critical phase of immune system development (1). Schools are high-density environments that can serve as sentinel sites, where children are frequently exposed to a diverse array of microbes, including both commensal and pathogenic organisms, allowing the study of these dynamics. This exposure can significantly influence the

Address correspondence to Jorge H. Valdes, jorge.valdes@gmail.com, or Claudia P. Saavedra, csaavedra@unab.cl.

Juan Castro-Severyn and Nicolás Pacheco contributed equally to this article. The author order was determined according to who participated the most in the analysis and was in charge of writing the manuscript.

The authors declare no conflict of interest.

See the funding table on p. 20.

composition and stability of the microbiome, making schools an important focal point for understanding the spread and evolution of microbial communities, especially in the context of emerging pathogens, antibiotic resistance, and global pandemics.

The human microbiome, particularly the communities of microorganisms residing in the nasopharynx and oral cavity, plays a crucial role in maintaining overall health and influencing disease susceptibility (2, 3). These microbial communities are involved in critical processes such as immune system development and regulation, protection against pathogens, and the maintenance of mucosal barriers (4–6). In children, the microbiome is highly dynamic, undergoing significant changes during early development, mainly influenced by diet, environmental exposures, interactions with caregivers and peers, and the progressive maturation of the immune system (7).

Understanding the composition and dynamics of the microbiome during childhood is essential for elucidating its potential role in health and disease progression. Previous reports have revealed associations between the nasopharyngeal microbiome composition and the risk of respiratory infections, including those caused by pathogens such as *Streptococcus pneumoniae* and *Haemophilus influenzae* (8, 9). The oral microbiome has been associated with dental caries and other oral diseases (10). In both cases, the microbiome not only reflects the immediate health status of the child but also serves as a potential predictive marker to assess future health outcomes (11, 12).

Advances in high-throughput sequencing technologies have enabled the detailed characterization of the human microbiome, revealing significant inter-individual variability as well as age-specific microbial signatures, and have also provided a global overview of microbiome assemblages (13). A study by Biesbroek et al. (14) showed that the nasopharyngeal microbiome in young children is highly diverse, with distinct microbial profiles that change over time. Similarly, a study by Lemon et al. (15) demonstrated significant differences in the bacterial communities present in the nostrils and oropharynx of children, underscoring the complexity of microbial ecosystems across different regions of the upper respiratory tract.

Several longitudinal studies have explored the dynamics of the pediatric microbiome, providing valuable insights into how these microbial communities evolve over time. One notable study by Bosch et al. (16) examined the development of the upper respiratory tract microbiota in infants and found that the mode of delivery (vaginal vs cesarean) and environmental factors (such as daycare attendance) significantly shaped the composition of the microbiome. Another study by Teo et al. (17) highlighted the impact of the infant nasopharyngeal microbiome on the severity of lower respiratory infections and the subsequent risk of developing asthma. These studies contribute to the understanding of microbial dynamics in early childhood, highlighting that early-life microbial exposures can exert long-lasting effects on health.

Despite the growing body of research on the pediatric microbiome, significant gaps remain in our understanding of the temporal stability and transitional dynamics of these microbial communities, particularly in relation to external factors such as the built environment children inhabit, seasonal fluctuations, and age-related physiological changes (18). Addressing this gap is key for developing strategies to prevent imbalances that may lead to disease and for informing public health policies (19). The microbiome acts as a first line of defense against pathogenic invasions, and alterations in its composition can influence infection susceptibility and the effectiveness of immune responses. Additionally, the microbiome's role in modulating the effects of antibiotics, both in terms of resistance and resilience, is critical (20). As antimicrobial resistance continues to rise, studies in this direction provide foundational knowledge for designing strategies for preserving and restoring beneficial microbial communities (21). This research can also contribute to the development of microbiome-targeted therapies and interventions aimed at preventing or mitigating the impact of emergent pathogens in pediatric populations.

In this study, we present a longitudinal analysis of the nasopharyngeal and oral microbiomes of children attending different grades in a public school, collecting samples

**TABLE 1** Distribution of subjects by gender, age, and grade[a]

| Characteristic | Value |
| --- | --- |
| Gender | |
| Female | 57 (47.9%) |
| Male | 62 (52.1%) |
| Age | |
| 4 yrs | 4 (3.4%) |
| 5 yrs | 12 (10.1%) |
| 6 yrs | 18 (15.1%) |
| 7 yrs | 12 (10.1%) |
| 8 yrs | 10 (8.4%) |
| 9 yrs | 16 (13.4%) |
| 10 yrs | 29 (24.4%) |
| 11 yrs | 5 (4.2%) |
| 12 yrs | 2 (10.1%) |
| 13 yrs | 1 (0.8%) |
| Grade | |
| Pre-kindergarten | 4 (3.4%) |
| Kindergarten | 14 (11.8%) |
| 1st grade | 16 (13.4%) |
| 2nd grade | 13 (10.9%) |
| 3rd grade | 12 (10.1%) |
| 4th grade | 15 (12.6%) |
| 5th grade | 30 (25.2%) |
| 6th grade | 7 (5.9%) |
| 7th grade | 8 (6.7%) |

[a]Total number of children enrolled in the study, $n = 119$. Total number of classrooms considered in the study, $n = 21$.

in autumn, winter, and spring. By clustering children based on their microbiome profiles at each time point, we identified distinct groups and evaluated the transitions that individuals underwent between these groups over time. This approach allowed us to gain insights into the temporal dynamics of the microbiome in a pediatric population, with potential implications for understanding the impact of environmental factors and microbial interactions on health outcomes.

## RESULTS

### Target school population

A total of 119 children were included in the study, with a nearly balanced distribution of boys and girls (47.9% females and 52.1% males). The age range of the sampled children was between 4 and 13 years old, with the largest group being 10-year-olds (24.4%). Sampling covered 21 classrooms across 9 grade levels (ages 4 to 13), with 5th-grade students (10 years old) the most represented individuals (25.2%). A total of 341 samples were collected across the three sampling campaigns, conducted in (April) autumn, (June) winter, and (September) spring of 2023, with an increasing number of samples collected in each successive campaign; spring yielded the highest number (153), nearly double that of autumn. Although we expected to retrieve both oral and nasopharyngeal samples from each participant in some cases, we only obtained one due to technical difficulties or participant discomfort, resulting in a total of 163 oral and 178 nasopharyngeal samples. Male participants (186) were more represented than female participants (155). This data set provides a well-distributed sample for studying temporal variations in the oral and nasopharyngeal microbiomes of schoolchildren, potentially influenced by the school built environment. The main characteristics of the sampled population are presented in Tables 1 to 3.

**TABLE 2** Distribution of subjects by age, based on anatomical site and season of sample collection

| Anatomical site and season | Age (yrs) | | | | |
|---|---|---|---|---|---|
| | Mean | SD | Min | Max | Median |
| Nasopharyngeal | | | | | |
| Autumn | 6.5 | 1.4 | 5 | 10 | 6 |
| Winter | 9.5 | 1.9 | 5 | 13 | 10 |
| Spring | 8.6 | 2.3 | 5 | 13 | 9 |
| Oral | | | | | |
| Autumn | 6.3 | 1.4 | 4 | 10 | 6 |
| Winter | 9.3 | 2.2 | 5 | 13 | 10 |
| Spring | 8.7 | 2.3 | 5 | 13 | 9 |

Sequencing of the 341 samples yielded an average of ~300,000 paired-end reads per sample. The alpha diversity of the nasopharyngeal and oral microbiomes was evaluated using the Chao1, Shannon, and Simpson indices, which together account for species richness, evenness, overall diversity, and dominance of the studied microbial communities. In summary, the values for the three indices were significantly higher in nasopharyngeal samples compared to oral samples (Mann Whitney test, $P < 0.0001$), indicating higher richness and diversity, but also increased dominance (Fig. 1). To address the possible variations in the communities, we compared the three sampling campaigns and found that alpha diversity remained relatively stable across seasons. For the oral samples, no significant variation was found in any index when comparing the samples from autumn, winter, or spring. Conversely, in the nasopharyngeal samples, we observed some significant differences in richness (Chao1) and evenness (Shannon) between samples from autumn compared to winter and spring. When comparing alpha diversity across different ages, no significant differences were observed in either the nasopharyngeal or oral microbiomes. However, slight trends were detected in younger children (6–7-year-olds), who exhibited lower microbial diversity in the nasopharynx compared to older children (5th and 6th graders, 10 and 11 years old), though these differences did not reach statistical significance. Moreover, although the mean ages of the sampled population in each season were variable, the analysis of variance and size effect (eta-squared) showed that age (or its interaction with season) explains only a small proportion of the observed variability in the microbiota.

## Beta diversity

Beta diversity was evaluated to assess differences in microbial community structure across samples. Principal coordinate analysis (PCoA) based on Bray-Curtis dissimilarity revealed no distinctive clustering of samples by anatomical site (nasopharyngeal vs oral) (Fig. 2), although nasopharyngeal samples exhibited a more concentrated clustering, indicating a more homogeneous community structure, while oral samples were more dispersed, reflecting higher variability. Permutational multivariate analysis of variance

**TABLE 3** Distribution of samples taken for sequencing

| Group | No. of samples | | | |
|---|---|---|---|---|
| | Total | Autumn | Winter | Spring |
| All samples | 341 | 74 | 114 | 153 |
| Female | 155 | 34 | 47 | 74 |
| Male | 186 | 40 | 67 | 79 |
| Oral | 163 | 34 | 59 | 70 |
| Nasopharyngeal | 178 | 40 | 55 | 83 |
| Female oral | 75 | 17 | 25 | 33 |
| Male oral | 88 | 17 | 34 | 37 |
| Female nasopharyngeal | 80 | 17 | 22 | 41 |
| Male nasopharyngeal | 98 | 23 | 33 | 42 |

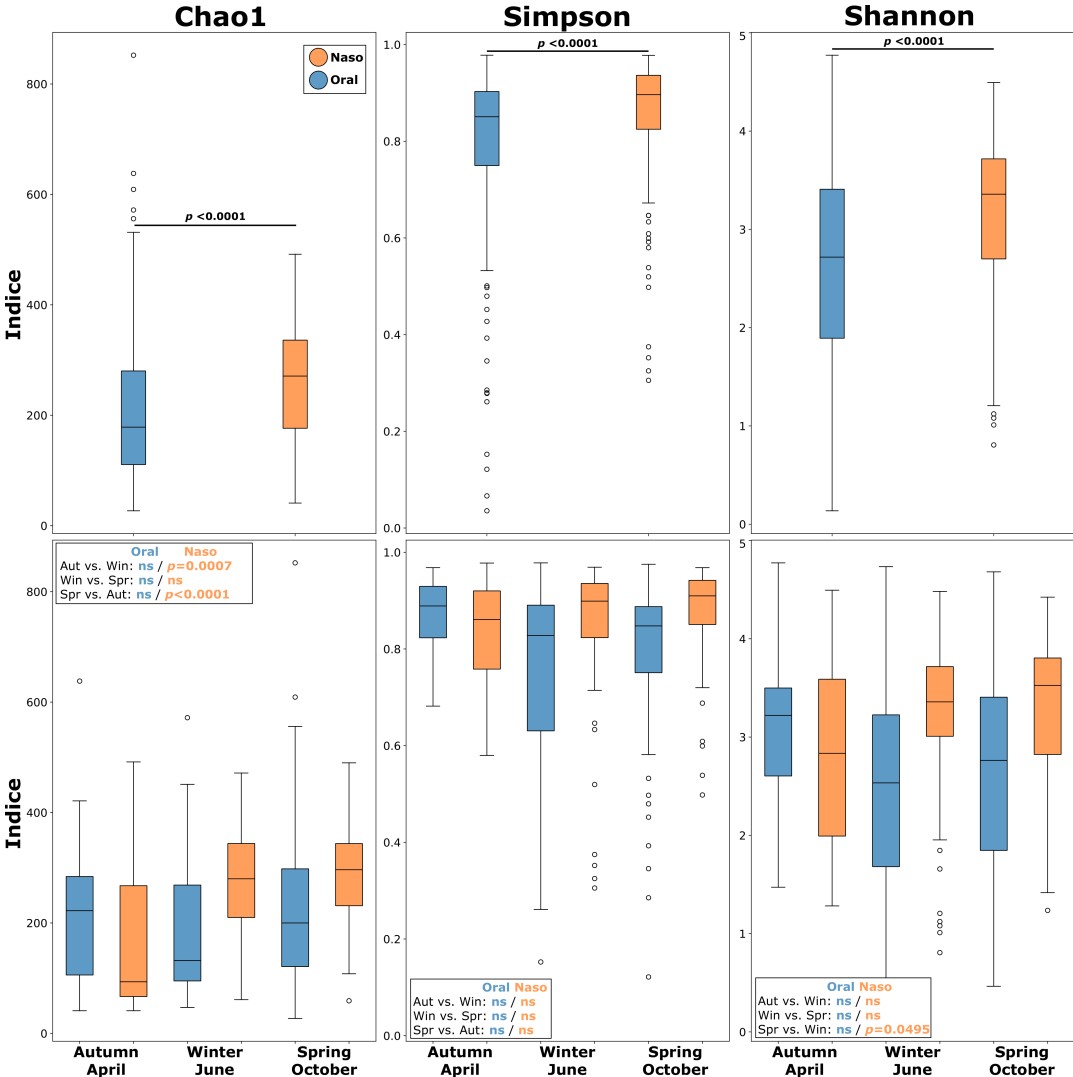

**FIG 1** Bacterial community diversity in oral and nasopharyngeal samples. Comparison of alpha diversity indices for all studied samples among anatomical sites and seasons. The boxes (colored according to anatomical site) represent the dispersion in the interquartile range, with the median indicated by the central line and the whiskers representing the range of values within 1.5 times the interquartile range; open circles indicate outliers beyond this range. Asterisks denote statistical significance ($P < 0.05$), while "ns" indicates non-significant differences.

(PERMANOVA) confirmed that sample type (nasopharyngeal vs oral) was a significant driver of microbial community structure ($R^2 = 0.45$, $P < 0.001$), consistent with the alpha diversity results.

Sampling season also contributed significantly to variation in microbiome composition, although to a lesser extent than sample type (PERMANOVA, $R^2 = 0.12$, $P = 0.03$). Although no clear segregation pattern was observed, the nasopharyngeal samples from spring tended to form a compact cluster, while those from autumn were more dispersed (Fig. 3). This pattern is less evident in the oral samples. Age group showed a minor and non-significant effect on beta diversity ($P > 0.05$).

### Relative abundance of key taxa

The taxonomic composition of microbial communities revealed high variability within the study population, which limits the ability to identify clear patterns between the anatomical sites and across seasons. Nonetheless, the most abundant taxa in all the samples included *Streptococcus, Moraxella, Proteus, Herbaspirillum, Haemophilus,*

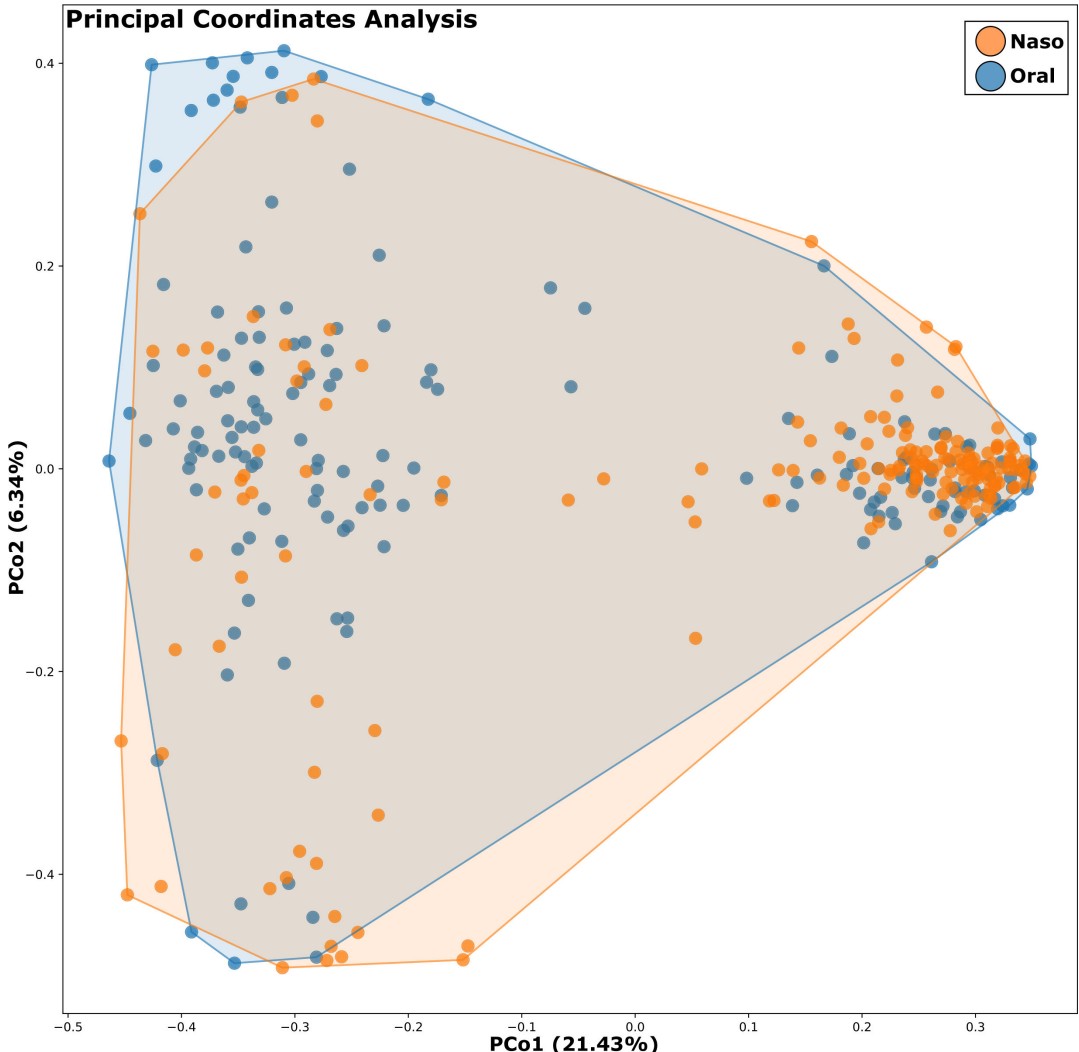

**FIG 2** Bacterial community segregation by anatomical site. PCoA based on amplicon sequence variants (ASVs) relative abundance, using Bray-Curtis dissimilarity as the distance metric; each point corresponds to a microbial community, colored according to anatomical site.

*Neisseria,* and *Staphylococcus,* but distinctive patterns were observed between the anatomical sites and seasons (Fig. 4).

To reduce the complexity of taxonomic composition analyses and pattern identification, we averaged sample data by season for each anatomical site and detected highly abundant genera, including *Streptococcus, Moraxella, Proteus, Neisseria,* and *Haemophilus* (Fig. 5), as previously described. Moreover, *Streptococcus* was found to be the most abundant (on average, 17% to 40%) among all samples, for both oral and nasopharyngeal sites. Additionally, a noticeable increase over time was observed, from 19% and 17% in autumn to 24% and 28% in winter, reaching 40% and 30% in spring, for oral and nasopharyngeal samples, respectively. In contrast, *Proteus* displayed the opposite trend, being highly enriched in autumn (27% and 24%) in both anatomical sites, declining significantly in winter (3% and 4%), and becoming undetectable by spring. A similar trend is observed for *Staphylococcus,* which was completely absent by spring, and *Kocuria,* which was only detected in autumn for both anatomical sites. Notably, *Moraxella* was particularly dominant in younger children, with a relative abundance of over 40% in 1st- and 2nd-grade students (6 and 7 years old), while *Streptococcus* showed higher prevalence in older children (5th- and 6th-grade students, 10 and 11 years old). Seasonal variation was observed; for example, the relative abundance of *Haemophilus* D increased

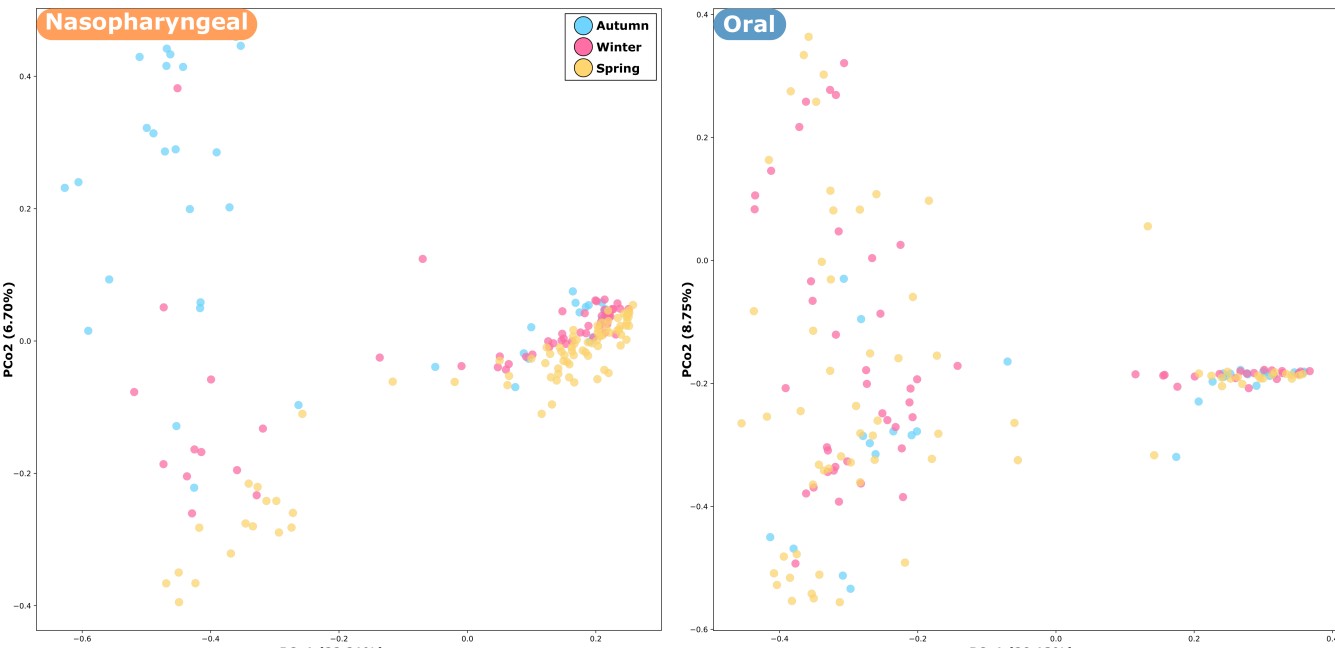

**FIG 3** Bacterial community segregation by season for both anatomical sites. PCoA based on ASV relative abundances, using Bray-Curtis dissimilarity as the distance metric; each point corresponds to a microbial community, colored according to season.

in spring compared to autumn and winter. Consistently, *Streptococcus* remained the most dominant genus, accounting for up to 40% and 30% of the community across seasons and age groups in both anatomical sites. Other genera, such as *Neisseria*, exhibited more pronounced temporal variation, particularly between winter and spring, when their relative abundance increased significantly in older children. The presence of *Prevotella* was also notable, especially in 3rd- and 4th-grade students (9 years old), where it accounted for approximately 15%–20% of the microbial community. The significance of all the different comparisons in the relative abundance of key genera was tested using the Wilcoxon rank-sum test.

By considering only the samples that showed tight clustering in the PCoA for both anatomical sites, we found a pattern of shared taxonomic composition (Fig. 6). These clusters consisted of 56 oral and 121 nasopharyngeal samples showing strong dominance of *Streptococcus* (~34%), *Neisseria* (~10%), and *Rothia* (~6%) genera. However, differences between both anatomical sites were found, *Haemophilus*-A being more abundant in oral samples. It is noteworthy that *Proteus* was detected in oral samples at a very low abundance (0.8%) and was not found in nasopharyngeal samples. In contrast, *Staphylococcus* was not detected in oral samples and showed very low abundance in nasopharyngeal samples (1%). Notably, the *Moraxella* genus was not detected in the clustered samples for both anatomical sites.

Samples in the nasopharyngeal cluster included a large proportion of children aged 9 to 10 years (47.1%), slightly more males (53.7%), and a predominance of samples from the spring sampling campaign. The majority (53.6%) of the oral sample cluster comprised 5–6-year-old children (51.7%), with an equal gender distribution and consistent representation across the seasons. Samples from only nine children were included in the clusters of both anatomical sites (samples A_S008, A_S017, A_S036, J_S036, J_S039, O_S003, O_S009, O_S016, and O_S048), representing 16.1% of the oral and 7.4% of nasopharyngeal samples.

Furthermore, to monitor the stability and changes in the children's microbiome over time, we selected samples from those subjects who were evaluated at all three seasons (autumn, winter, and spring). Only 12 (10.1%) of the 119 enrolled children participated in all three sampling campaigns, seven for oral samples and nine for

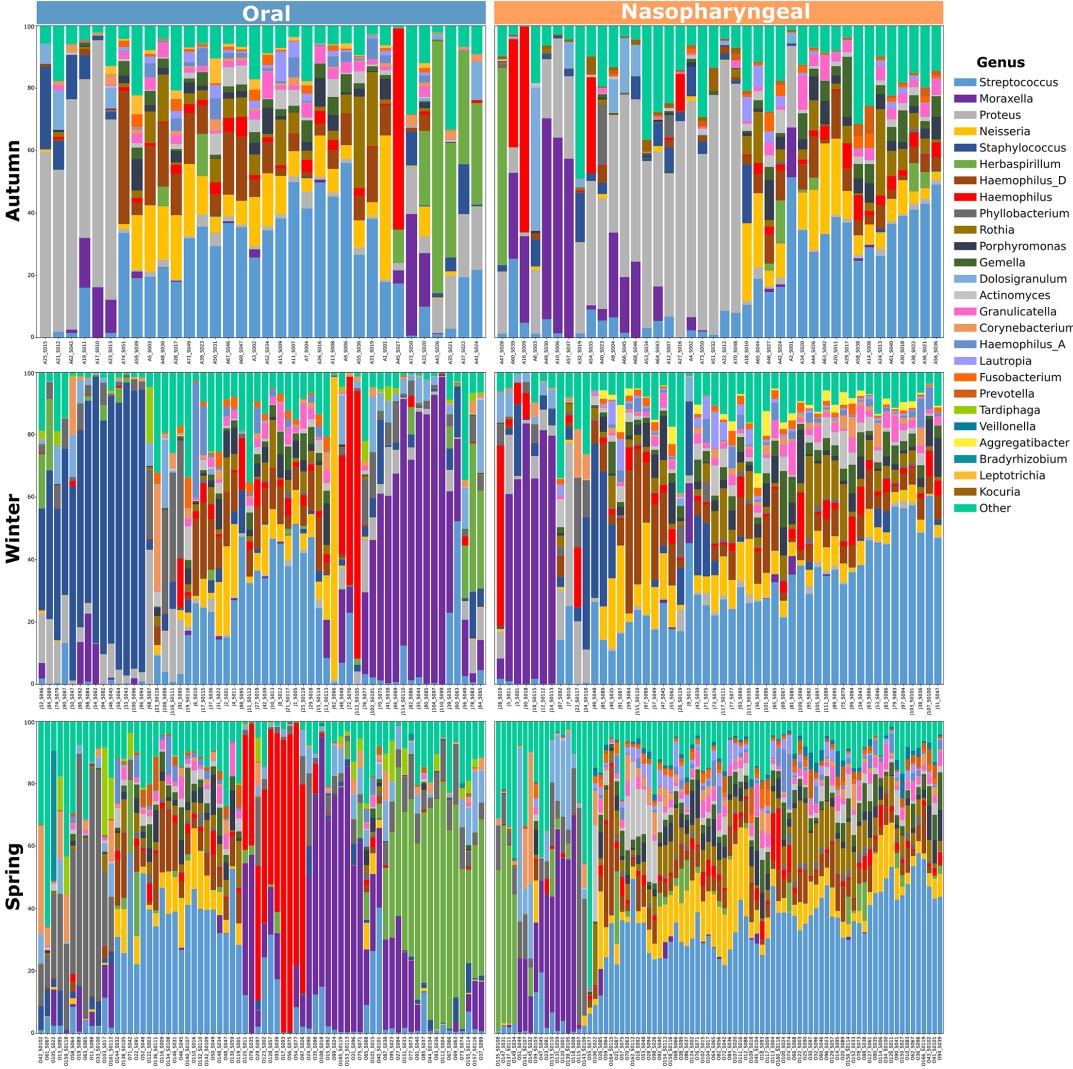

**FIG 4** Taxonomic composition of the studied microbial communities. Relative abundance of the top 26 genera across the seasons for both anatomical sites. Taxa are color-coded by genus, and less abundant genera are grouped under the "Other" category.

nasopharyngeal samples (four in both anatomical sites). The analysis revealed distinct taxonomic compositions between the oral and nasopharyngeal microbiomes, with notable differences in the abundance of *Proteus, Moraxella,* and *Neisseria* genera across the volunteers and time points (Fig. 7), despite *Streptococcus,* which generally remains as the dominant genus. In this case, the alpha diversity indices (Shannon and Simpson) were higher in the oral microbiome compared to the nasopharyngeal samples, which contrasts with what was observed when considering the complete data sets.

Temporal fluctuations were observed, as microbial composition and diversity varied between autumn, winter, and spring. The *Proteus* genus was only present in samples collected in autumn and winter, mostly in nasopharyngeal samples from autumn (S045, S046, S048). In oral samples, *Moraxella* was rarely detected, in contrast to its presence in nasopharyngeal samples (detected in six of the nine evaluated children). Moreover, *Neisseria* is present in the samples where *Moraxella* was absent or present in very low abundance (both for oral and nasopharyngeal). *Staphylococcus* also appeared in higher proportions in some oral (S047) and nasopharyngeal (S048 and S001) samples from winter. In addition, *Prevotella* was exclusively detected on the nasopharyngeal samples, and *Tardiphaga* only in oral samples. Notably, some subjects exhibit more pronounced shifts in diversity and taxonomic composition, showing highly different patterns across

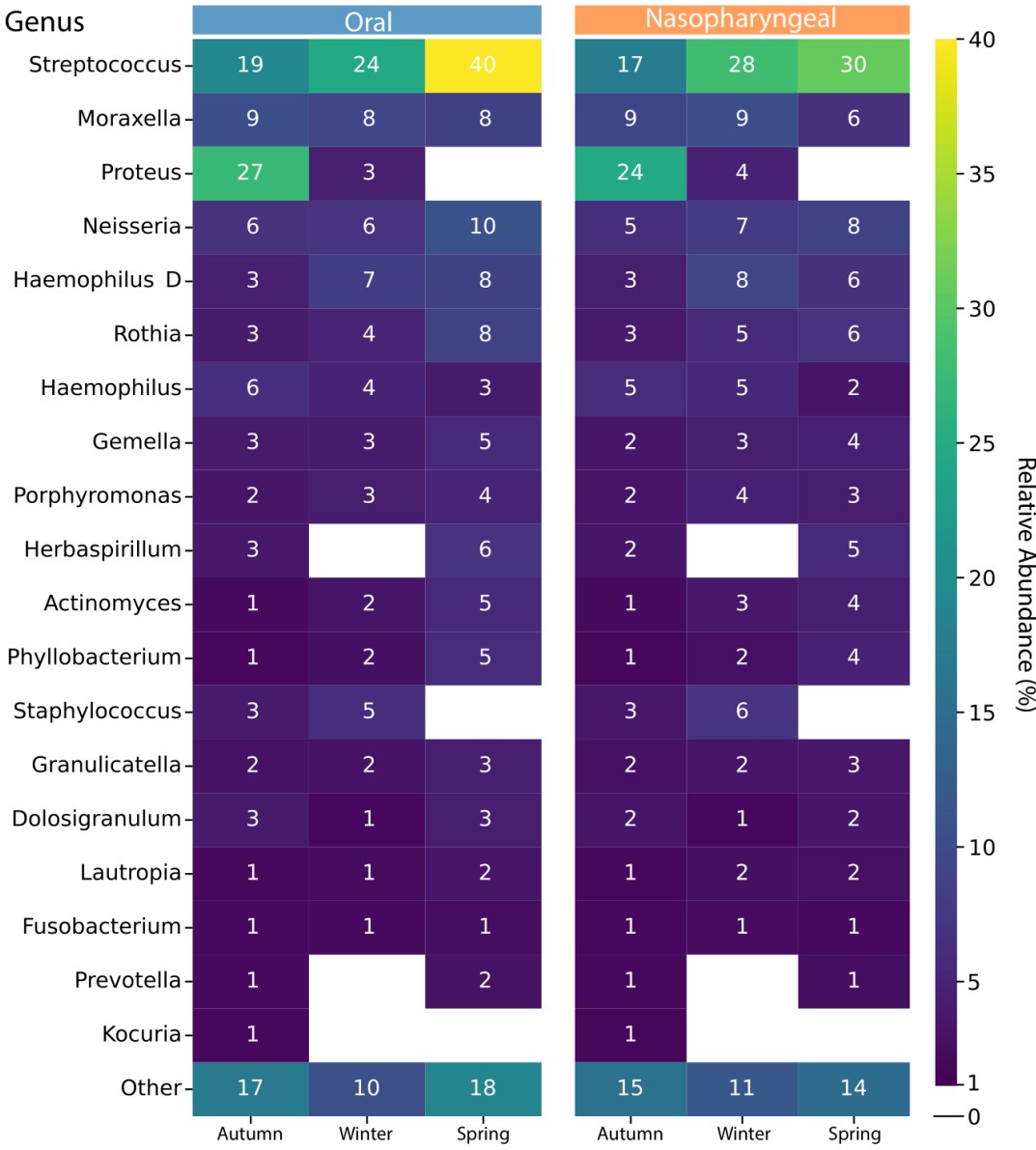

**FIG 5** Most abundant genera in the microbial communities. Heatmap showing the 20 most abundant genera averaged by season for both anatomical sites. The color gradient and the numerical values inside each cell indicate the relative abundance (in percent) for each genus.

the seasons for oral (S036, S022, and S011) and nasopharyngeal (S045, S048, S035, and S001) samples. While in most cases, roughly similar profiles are observed at two time points, the third one displays an imbalanced profile, suggesting a possible dysbiotic shift.

## Clustering analysis

Hierarchical clustering of microbiome profiles identified distinct groups of children based on their microbial community composition for all time points. For the nasopharyngeal microbiome, three major clusters consistently emerged across the study period (autumn to spring) (Fig. 8). Each cluster was characterized by a dominant set of taxa, with cluster N-XIII being the largest and enriched in *Streptococcus*, followed by clusters N-IV and N-V, enriched in *Moraxella* and *Proteus* genera. In smaller proportions, clusters N-I and N-II/N-XI displayed a significant abundance of *Herbaspirillum* and *Haemophilus*. In the oral microbiome, while cluster O-V (enriched in *Streptococcus*) was the largest, the clustering was more varied compared to nasopharyngeal samples (Fig. 9). The following

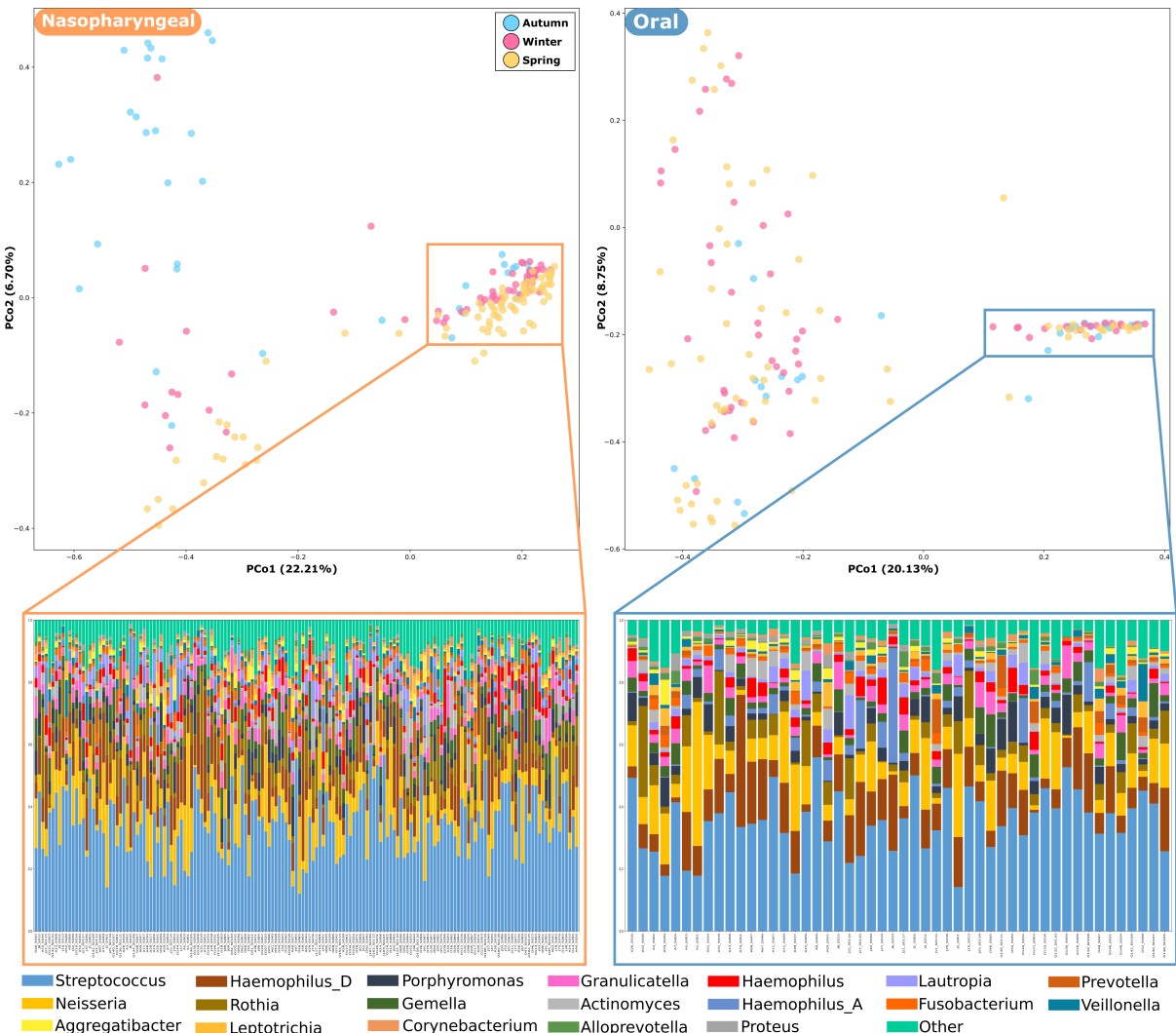

**FIG 6** Taxonomic composition of samples forming tight clusters identified in the PCoA plots for both anatomical sites. The taxonomic composition of those clusters is displayed as the relative abundance of the top 29 genera. The stacked bar compositions were hierarchically clustered according to their taxonomic profile similarities, color-coded by genus. Less abundant genera are grouped under the "Other" category.

two clusters, O-VIII and O-IX, were found to be enriched in *Moraxella* and *Herbaspirillum*, respectively. In addition, we identified other smaller clusters enriched in *Staphylococcus* (O-I), *Phyllobacterium* (O-IV), *Haemophilus* (O-VII), and *Proteus* (O-II). The statistical significance of enriched genera was determined for each cluster by comparing the abundance against all others composing the microbial communities within that cluster (Fig. S1).

## Transition network analysis

Transition analysis revealed that a large proportion of children remained in the same cluster over time, particularly those in cluster N-XIII (enriched in *Streptococcus*) (Fig. S1). However, a significant proportion of children transitioned between clusters, indicating shifts in the dominant taxa, with these transitions being most pronounced between autumn and winter. To identify statistically significant movements, Fisher's exact test was applied to all observed transitions. For the nasopharyngeal microbiome (autumn–winter), notable significant transitions ($P < 0.05$) included the shift from cluster N-XIII to cluster N-IV (number of transitions = 152; odds ratio = 24; $P$-value = 0.006434) and for the nasopharyngeal microbiome (winter–spring), another notable significant transition

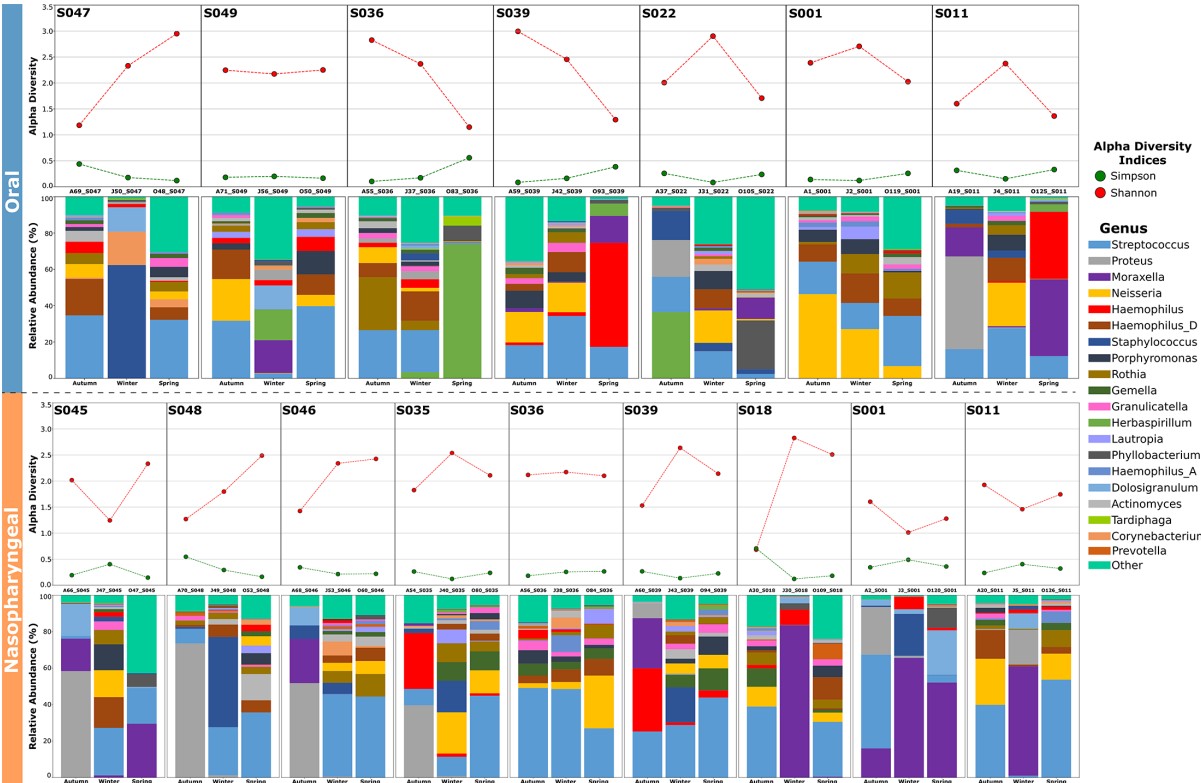

**FIG 7** Taxonomic composition profiles of the participants who were sampled in the three seasons: seven oral (top) and nine nasopharyngeal (bottom) samples. For each subject, the stacked bar plots represent the relative abundance of bacterial genera in each sample (color-coded by genus), and the dot plots represent the values of Shannon and Simpson diversity indices.

included the shift from cluster N-XII to cluster N-III (number of transitions = 209; odds ratio = 104; *P*-value = 0.00283). Other significant transitions are visually highlighted in Fig. 10. Transitions were more frequent in the oral microbiome, with children's microbiome composition often transitioning between the identified clusters, especially from autumn to spring, highlighting more dynamic behaviors in the oral microbiome.

To further explore the interactions between bacterial taxa, transition networks were constructed for both nasopharyngeal and oral microbiomes (Fig. 10). In the nasopharyngeal transition network, *Streptococcus* was identified as a key hub genus, showing positive association with *Moraxella* but a negative association with *Staphylococcus* and *Neisseria*. The transition network structure suggests that *Streptococcus* might play a central role in shaping the nasopharyngeal microbial community, potentially influencing the relative abundance of other taxa. In the oral transition network, *Streptococcus* was also the dominant hub genus, showing positive associations with *Phyllobacterium* and *Herbaspirillum* and negative associations with *Proteus*. This suggests that *Streptococcus* may exert competitive pressure on other members of the oral microbiome, potentially influencing community composition and stability over time. Overall, these transition network analyses highlight key microbial interactions that may contribute to the stability and resilience of the nasopharyngeal and oral microbiomes in children.

## Summary of findings

- Alpha diversity was significantly higher in nasopharyngeal samples compared to oral samples, with low overall temporal variability.

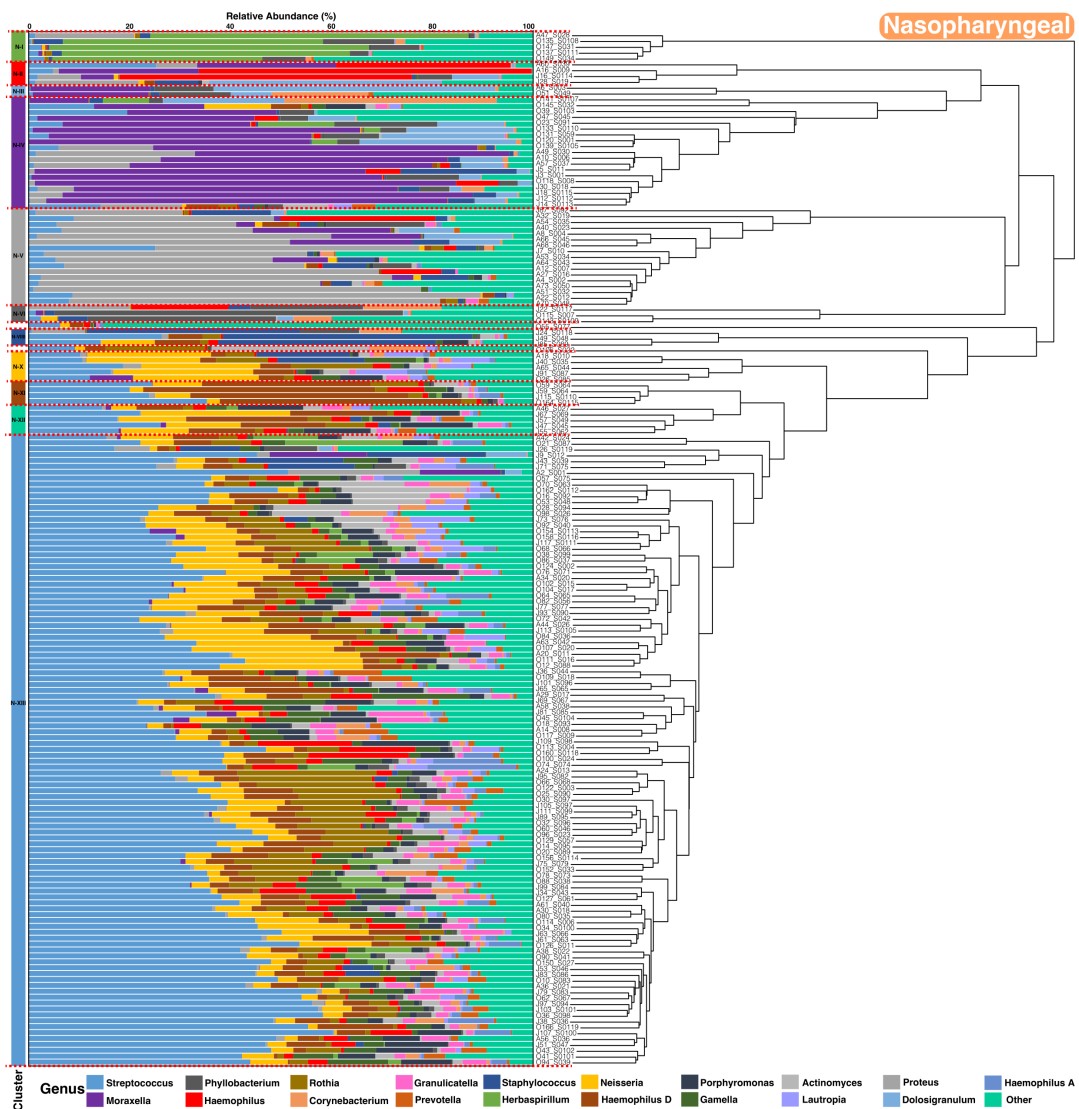

**FIG 8** Dendrograms depicting sample relatedness based on hierarchical clustering of taxonomic composition patterns, derived from the relative abundance of ASVs in nasopharyngeal samples collected during autumn, winter, and spring. Taxa are color-coded by genus, and less abundant genera are grouped under the "Other" category. On the left, clusters are numbered I to XIII and colored according to the dominant genus.

- Beta diversity, although revealing a similar pattern between the two anatomical sites, showed more densely clustered nasopharyngeal microbiomes, suggesting a potentially more stable structure.
- Hierarchical clustering detected 11 clusters in the nasopharyngeal microbiome and nine in the oral microbiome, with significant transitions between clusters over time, the largest of which was enriched in *Streptococcus*.
- Relative abundance analyses revealed that *Streptococcus* was the most abundant genus in both anatomical sites, followed by less prevalent *Moraxella*, while *Proteus*, *Neisseria*, and *Haemophilus*. The most important changes identified among all analyzed samples were mainly driven by fluctuations in the abundance of these genera.
- Beta diversity analysis further demonstrated that taxonomic composition profiles of samples forming compact clusters were highly similar—both within and between anatomical sites—characterized by a high proportion of *Streptococ-*

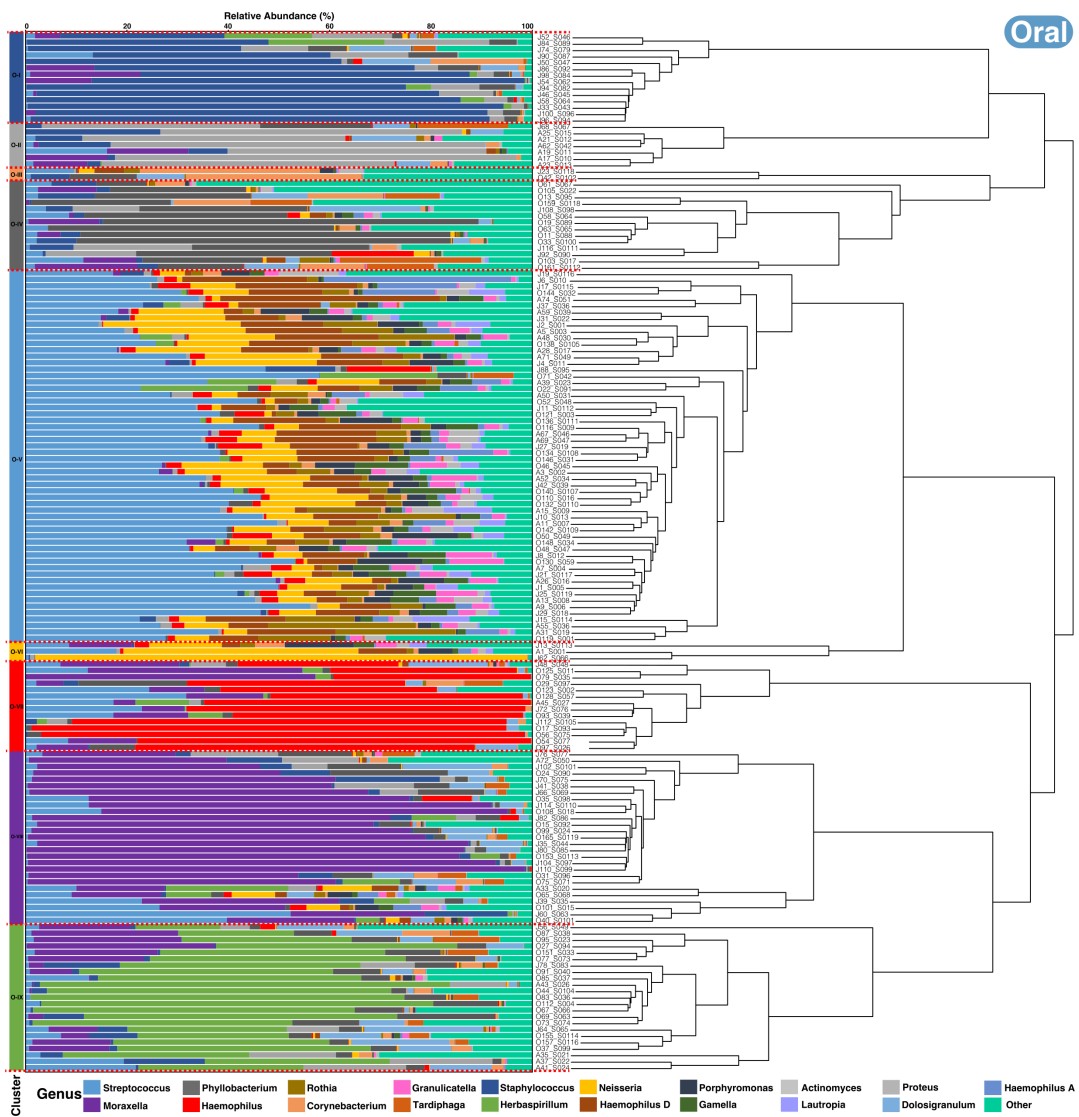

**FIG 9** Dendrograms depicting sample relatedness based on hierarchical clustering of taxonomic composition patterns, derived from the relative abundance of ASVs in oral samples collected during autumn, winter, and spring. Taxa are color-coded by genus, and less abundant genera are grouped under the "Other" category. On the left, clusters are numbered I to IX and colored according to the dominant genus.

*cus* and *Neisseria*, accompanied by low abundance of *Proteus* and absence of *Moraxella*.

- Transition network analysis identified *Streptococcus* as the key hub genus, indicating potential associations with *Phyllobacterium* and *Herbaspirillum* in the mouth, and, with *Moraxella*, *Staphylococcus,* and *Neisseria* in the nasopharynx, highlighting their potential influence on microbial community structure and maintenance.

- These findings underscore the dynamic nature of the pediatric microbiome and offer novel insights into the factors that drive fluctuations in microbial composition over time and across different anatomical sites.

## DISCUSSION

One of the goals of the "School MicroBE" initiative is to characterize the temporal dynamics and community shifts in nasopharyngeal and oral microbiomes of 119 children aged 4 to 13 years attending a public school in Chile. Accordingly, alpha diversity indices

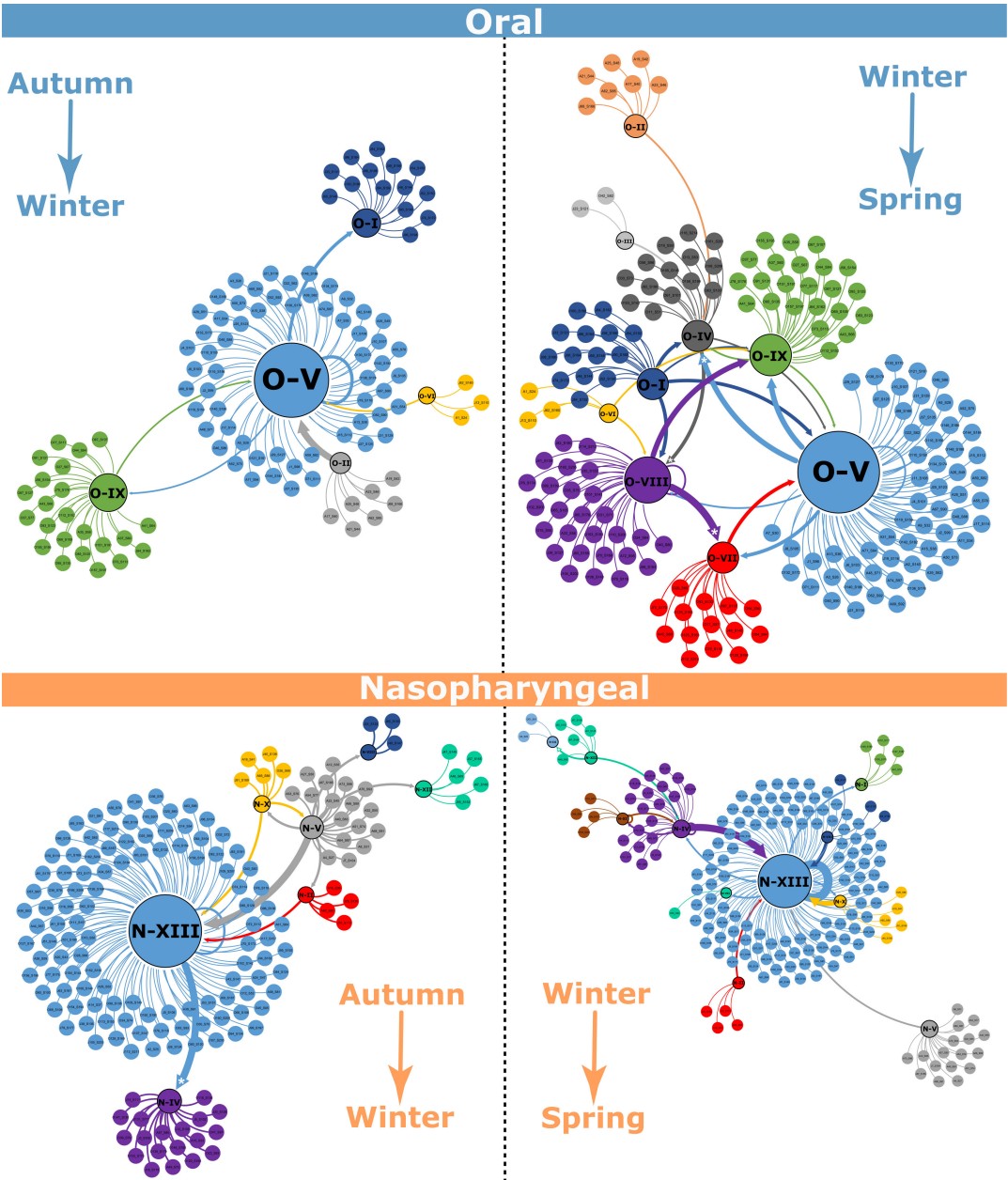

**FIG 10** Transition networks for oral and nasopharyngeal samples from autumn to winter and from winter to spring. Node colors represent distinct clusters (previously defined). Central nodes correspond to groups of samples, with their size proportional to the number of samples assigned to each group in the respective seasonal comparison. Arrows indicate transitions of individual samples from one group to another group between seasons. The arrow width reflects the number of samples transitioning, and the arrow tip reflects the transition direction. The five asterisks on the arrow tips denote statistically significant transitions.

showed that (on average) the nasopharyngeal microbiome was more diverse than the oral microbiome across the three seasons, although it is not the most common; it has been reported before (22). Moreover, nasopharyngeal microbiome diversity varied significantly between seasons, being significantly lower in autumn, compared to winter and spring. This could be associated with several different factors, indicating that the nasopharyngeal microbiome composition is strongly influenced by season, age, and social interactions, while the oral microbiome is more influenced by lifestyle-related variables such as diet, tobacco, alcohol, and antibiotics usage (23). This evidence explains

in part why the oral microbiome could be more dynamic compared to the nasopharyngeal, reflecting a higher number of transitions among the oral samples. Moreover, the low microbial diversity in autumn may be associated with the high incidence of seasonal allergic rhinitis following summer, which is well known to have a strong impact on the composition and structure of the nasal microbiomes (24). The alterations that viral respiratory infections can cause on the nasal and oral microbiomes are also well known (25, 26). This is particularly relevant in Chile, where the relaxation of the COVID-19 pandemic restrictions in 2021 led to a surge in respiratory virus infections exceeding pre-pandemic levels (27).

Nasopharyngeal microbiomes appear to be more stable over time, despite being more diverse, which agrees with previous reports suggesting that these communities undergo changes related to age or life stages (28, 29). These changes in diversity over time not only occur in healthy children but also in the nasopharyngeal microbiome of children with asthma (18). On the other hand, we observed a repeated pattern in samples taken from both anatomical sites, where a significant number of samples clustered together based on a common taxonomic composition, but some were more variable. We hypothesize that this group with a "more-stable" community could be a characteristic feature of healthy children, dominated by *Streptococcus* spp. and *Neisseria* spp.; while the others who deviate from this group could have been perturbed by allergic events, active illness, or recent infection, in those samples, *Moraxella* spp. was detected. Microbiome perturbations have been described for children with adenoids, otitis media, and lower respiratory tract infections (30, 31). Lower respiratory tract infections were associated with a high prevalence of *Haemophilus influenzae*, *Klebsiella* spp., and *Streptococcus pneumoniae*, while *Fusobacterium* spp. and *Peptostreptococcus* spp. were enriched in children with otitis media.

Our results show that *Proteus* abundance decreased across seasons: in autumn, it represented 27%–24% of the microbiota for both anatomical sites, but by winter, the abundance dropped significantly, and it was not detected in spring. We must consider the effect of climatic changes that occur between seasons on the microbiome composition, where the habits and diet of the population change, and they also face allergies and seasonal respiratory viruses. Moreover, transition network analyses indicated that children from the group with abundant *Proteus* transitioned to a *Neisseria*-dominated group, particularly between autumn and winter, which corresponds to the most significant decline in *Proteus* abundance. Members of this genus have been widely recognized as a human pathogen due to their large repertoire of virulence factors, such as production of fimbriae, urease, hemolysins, metallophores, biofilm formation, and resistance to extended-spectrum β-lactamases and carbapenemases (32). In addition, gastrointestinal, lung, urinary tract, and bloodstream infections, rheumatoid arthritis, and Crohn's disease have been reported to be caused by *Proteus* species (*P. mirabilis, P. vulgaris, P. hauseri, and P. penneri*) (33, 34); however, we found no reports linking *Proteus* with the oral or nasopharyngeal microbiomes.

Another notable finding was the detection of the genus *Prevotella* exclusively in the nasopharyngeal microbiomes, while the genus *Tardiphaga* was detected only in the oral microbiomes. The prevalence of *Prevotella* in the nasopharynx is known, and it has also been reported that during lower respiratory tract infections, the abundance of this genus decreases while the abundance of *Moraxella* increases (35). *Tardiphaga* detection in oral microbiota samples was unexpected, as there is limited information available regarding this genus. Existing reports associate this bacterium with environmental soil samples or plant roots (36, 37); this could be due to the children's constant contact with soil during playground time. We did not find reports associating *Tardiphaga* with the microbiomes of either anatomical site evaluated.

The information provided supports that taxa responsible for maintaining nasopharyngeal microbiome structure (key hub genera) are *Streptococcus*, *Haemophilus*, and *Moraxella*, whose transition network suggests potential interactions both among themselves and with other taxa that might support the overall community structure.

The relevance of their role has been extensively addressed by multiple research groups (38), both in healthy and ill individuals (23, 31). *Streptococcus* spp. was by far the most dominant taxon, which is not surprising as this bacterium has been previously associated with these anatomical sites and diverse taxonomic groups. *Streptococcus oralis* and *Streptococcus salivarius,* which are known to be common commensals in these anatomical sites, can inhibit pathogen growth and biofilm formation (e.g., *Streptococcus pyogenes* and *Candida albicans*) due to bacteriocin production (39, 40). Additionally, these commensals form their own biofilms acting as protective barriers on epithelial cells, reducing the adherence and internalization of opportunistic pathogens, and thereby mitigating their cytotoxic effects (41).

Our results support strong associations between *Streptococcus* spp. and *Moraxella*, as well as *Staphylococcus* with *Neisseria* in the nasopharyngeal microbiome, while in the oral microbiome, associations were observed between *Phyllobacterium* and *Herbaspirillum*. In the case of *Neisseria*, this genus is recurrently found in the respiratory tract and has also been reported to be enriched in children affected by severe acute respiratory syndrome coronavirus 2 infection compared to healthy controls (42, 43). Additionally, a previous study that monitored the infant microbiome in South Africa from birth to 30 months found that *Neisseria* was one of the most variable taxa across seasons and was influenced by other factors, such as antibiotic use (30). The *Phyllobacterium* genus has been detected in the oral microbiome; notably, its abundance was reported to decrease in patients with systemic lupus erythematosus as disease activity decreased, compared to a control group (44). This genus has also been described as an atypical pathogen, previously detected in children with cystic fibrosis (45) and in children infected with Influenza A, but undetected in healthy children (46). One study in Chile characterized the nasal microbiome of 110 healthy young adults and found that the *Phyllobacterium* genus was among the most prevalent taxa, detected exclusively in the nasal microbiomes of women (47). *Herbaspirillum* species are predominantly recognized as plant-associated bacteria and are not commonly identified within the human oral microbiome (48). However, in recent years, this genus has been detected in case reports of human infections and associated with pathologies such as gastric and colorectal cancer (49–51).

Bacteria belonging to the *Haemophilus* genus are characterized as opportunistic and pathogenic organisms, frequently colonizing the upper respiratory tract. Many of these species have been targets of therapeutic efforts and vaccination strategies (31). However, in recent years, specific strains and serotypes involved in recurrent respiratory infections have been described (52). Although the *Moraxella* genus is frequently associated with the nasal microbiome, our results revealed a significant abundance in both anatomical sites. A study evaluating the nasopharyngeal microbiome of children aged 4 to 7 years identified *Moraxella* as the dominant taxon associated with an increased risk of upper respiratory infections and sinusitis, suggesting that its prevalence in the nasopharynx may influence susceptibility to these conditions (53). Conversely, research comparing the nasal and oropharyngeal microbiota of elderly individuals with respiratory tract infections and healthy controls found that *Moraxella* was less prevalent in patients with lower respiratory tract infections, highlighting its potential protective role in older adults (54). In addition, several reports have associated *Streptococcus*, *Haemophilus*, and *Moraxella* with respiratory virus infections, particularly rhinoviruses and adenoviruses (55, 56). Also, in a work where the microbiome of 90 nasal and 100 nasopharyngeal samples from adults were clustered, the authors found different types of communities, each dominated by members of the *Moraxella, Streptococcus, Neisseria, Haemophilus, Staphylococcus, Corynebacterium, Dolosigranulum, Fusobacterium,* and *Alloprevotella* genera, being the first seven coincident with those identified in this work (57). Based on the evidence presented, *Moraxella* was found in significant abundance in samples that deviated from the common pattern, potentially reflecting recent infection. Overall, these findings underscore the dynamic nature of the microbiome in both anatomical sites, emphasizing site-specific ecological patterns and seasonal variability. The predominant genera fluctuate over time (autumn → winter → spring), reflecting shifts in microbial

community composition. Certain genera reach high relative abundances (up to 40%) and vary seasonally, indicating potential environmental or host-related influences.

Schools are high-density environments where children are frequently exposed to a diverse array of microbes. Our results provide novel insights into the temporal dynamics and structural organization of the nasopharyngeal and oral microbiomes in school-aged children, suggesting complex ecological interactions and their potential associations with environmental factors, including host physiology and external exposures such as school activities and seasonal illnesses. The increase in alpha diversity observed during the warmer months suggests that the nasopharyngeal microbiome in children may undergo significant expansion and diversification during periods when they spend more time outdoors. Additionally, the identification of both synergistic and competitive interactions among microbial taxa suggests that microbiome stability is maintained through a balance of these relationships. For example, the positive association between *Streptococcus* and *Herbaspirillum* could indicate a mutually beneficial interaction that contributes to the stabilization of the nasopharyngeal microbiome during specific periods. Conversely, the fragmented networks observed in more diverse clusters may reflect a competitive environment in which no single genus dominates, leading to a more dynamic and potentially less stable microbial community.

Understanding the dynamics of the pediatric microbiome is crucial for identifying baseline compositions and deviations associated with emerging pathogens and antimicrobial resistance. While this study provides valuable insights, several limitations should be acknowledged, including the fact that only a single public school was sampled, and the findings may not be generalizable to other populations with different environmental, dietary, or socio-economic conditions. Additionally, while we focused on bacterial communities, future studies should consider the role of other microorganisms, such as viruses and fungi, in shaping the pediatric microbiome. Finally, a more extended longitudinal study spanning multiple years would provide a deeper understanding of the long-term stability and evolution of the microbiome in children within built environments. These studies could support the development of microbiome-based interventions, such as probiotics or microbiome-targeted therapies, aimed at enhancing children's natural defenses against pathogens—particularly in the southern hemisphere, where data remain limited. By promoting a healthy microbiome, such interventions could reduce the impact of infectious diseases in school-aged populations, ultimately contributing to more resilient communities in the face of emerging public health threats.

## Conclusion

This study presents a comprehensive analysis of the nasopharyngeal and oral microbiomes in school-aged children, revealing significant seasonal and temporal dynamics in microbial diversity and community composition. Our findings highlight the importance of considering temporal factors in microbiome research and emphasize the need for further studies to explore the potential mechanisms driving these changes. Understanding the factors that influence microbiome stability and transitions over time will be critical for developing strategies to promote and maintain healthy microbial communities in children, thereby supporting overall health and providing foundational evidence to assess the impact of disease during childhood. These results underscore the importance of temporal dynamics in elucidating the role of the microbiome in pediatric health and disease, offering a valuable baseline for future research aimed at optimizing child health through microbiome-targeted interventions.

## MATERIALS AND METHODS

### Study design and population

This research was conducted as a longitudinal observational study in the "Rebeca Matte" public school located in Renca, Santiago de Chile, Chile. The study population consisted

of children aged 4 to 13 years. Children with chronic respiratory conditions (asthma and chronic obstructive pulmonary disease), those undergoing antibiotic treatment within 3 months prior to sample collection, or those with known immunodeficiencies were excluded from the study.

## Sample collection

Nasopharyngeal and oral samples were collected from each participant at three time points: autumn (April), winter (June), and spring (October), corresponding to months 2, 4, and 8 of the school year in the southern hemisphere, excluding the summer break when children do not attend school. The climatic conditions of each season are contrasting: autumn (March to June), average temperature minimum of 13°C and maximum of 29°C, humidity of 70%–80%, and rainfall of 10–60 mm; winter (June to September), average temperature minimum of 3°C and maximum of 16°C, humidity of 50%–60%, and rainfall of 50–100 mm; and spring (September to December), average temperature minimum of 8°C and maximum of 25°C, humidity of 50%–60%, and rainfall of 10–30 mm (data from the Meteorological Direction of Chile). These conditions bring changes in available food, outdoor activities, as well as seasonal illnesses and allergies. Sample collection was conducted during school hours by trained healthcare professionals following standardized procedures. All samples were collected using sterile flocked swabs (BOEN Healthcare, Jiangsu, China), which were immediately placed into sterile tubes containing Viral Transport Medium (AllTest Biotech, Hangzhou, China). For nasopharyngeal samples, the swab was inserted into the nasopharynx and gently rotated; for the oral samples, the swab was gently rubbed along the inner cheeks, gums, tongue, and palate for approximately 30 seconds. The samples were taken just before the children attended their breakfast break. There was no other intervention, such as rinsing, apart from just brushing their teeth when they woke up at home. All samples were stored on ice and transported to the laboratory within 2 hours of collection.

## DNA extraction

Total DNA was extracted immediately after samples were taken, using 200 uL from the nasopharyngeal and oral samples with the automated nucleic acid isolation system EXM3000 (Zybio, Chongqing, China) with the Nucleic Acid Extraction Kit based on magnetic beads (Zybio, Chongqing, China) following the manufacturer's instructions. Blank and environmental samples were also used for DNA extraction to test the supplies, reagents, and any source of foreign DNA. The remaining samples were stored at −80°C. DNA quality and concentration were measured using a NanoDrop spectrophotometer (Thermo Fisher Scientific, USA).

## 16S rRNA amplicon sequencing

DNA samples were sent to SeqCenter (Pittsburgh, PA, USA), for the amplification of bacterial/archaeal 16S rRNA gene (V3-V4 region ~450 bp) using the primers 341F (CCTAYGGGGYGCWGCAG) and 805R (GACTACNVGGGTMTCTAATCC); the construction of $2 \times 300$ bp paired-end libraries using the Quick-16S Plus NGS Library Prep Kit (Zymo Research, Irvine, CA, USA) and the sequencing on a NextSeq 2000 platform (Illumina, San Diego, CA, USA). Negative controls (no DNA template) were included in each sequencing run to monitor for potential contamination.

## Bioinformatic analysis

Raw sequencing reads were processed using the DADA2 pipeline v2.26 (58) in R version 4.3.0 (59) to infer amplicon sequence variants (ASV) for each sample. Briefly, reads were filtered for quality (truncLen=c(290,230), maxN=0, maxEE=c(2,5), truncQ=2, minLen=50), after removing primers and adapters using the Cutadapt (60). Sequence variants were inferred after denoising, and chimeric sequences were removed. Following, an ASV table

was built to allow a maximum of two to five expected errors, removing chimeras, and the taxonomy table was built by assigning the classification using the formatted Genome Taxonomy Database (GTDB v207) (61). Any ASVs classified as chloroplasts or mitochondria were removed from the data set. To compare microbial diversity and community composition across samples, the Phyloseq v1.50 package (62) was used to determine alpha diversity, considering the Shannon and Simpson alpha diversity measures. Beta diversity of all samples in the data set was calculated through Bray-Curtis dissimilarity via the vegan v2.6-4 package (63). Finally, the relative abundance for each sample at the phylum, order, family, and genus ranks was obtained. PCoA was performed to visualize differences in community structure across time points and sample types.

## Clustering and transition analysis

Children were clustered according to their microbiome profiles at each time point using hierarchical clustering with Ward's method implemented via the pvclust v2.2-0 package (64). The number of clusters was determined by assessing the silhouette scores and inspecting the resulting dendrograms. A transition analysis was performed to evaluate the movement of individuals between clusters over the study period. Network diagrams were generated using Gephi v0.10.0 to (65) visualize these transitions.

## Statistical analysis

All statistical analyses were performed using R version 4.3.0 (59). The statistical significance of differences between groups was assessed using PERMANOVA. Differences in microbial diversity metrics between groups were assessed using the Wilcoxon rank-sum test for pairwise comparisons and the Kruskal-Wallis test for comparisons across multiple groups. Additionally, multiple independent $t$-tests were performed to identify the dominant taxa in each of the groups revealed by the hierarchical clustering. For transition analysis, the significance of cluster transitions was evaluated using Fisher's exact test. All $P$-values were adjusted for multiple comparisons using the Benjamini-Hochberg procedure, with a significance threshold set at $\alpha = 0.05$.

## ACKNOWLEDGMENTS

We thank Universidad Central's computing cluster for providing data storage, support, and computing power for bioinformatic analyses. We thank the entire Escuela Rebeca Matte Bello community (students, teachers, workers, and board) for their collaboration, participation, and initiative to be part of a scientific study, especially the children's curiosity, interest, and willingness, which was a great motivational force.

This research was sponsored by grants from ANID (Chilean National Research and Development Agency), mainly by the ANILLO ATE220007 and the Regular FONDECYT 1250419 to C.P.S. The sponsors and financing agencies had no role in the study design, data collection and analysis, the decision to publish, or the preparation of the manuscript.

J.C.-S.: formal analysis, methodology, original draft writing, review, and editing. N.P.: conceptualization, investigation, methodology. G.V.: conceptualization, investigation, methodology, visualization, original draft writing, formal analysis, data curation. G.I.K.: conceptualization, investigation, methodology, visualization, formal analysis, data curation. C.P.-E.: formal analysis, review, and editing. F.R.: review. A.G.: review. G.A.: review and editing. F.V.-E.: review. J.O.-P.: review and editing. J.H.V.: conceptualization, investigation, formal analysis, data curation, funding acquisition, resources, supervision, writing original draft, review, and editing. C.P.S.: conceptualization, formal analysis, funding acquisition, project administration, resources, supervision, writing, review, and editing.

The authors declare that the research was conducted in the absence of any commercial or financial relationships that could be construed as a potential conflict of interest.

## AUTHOR AFFILIATIONS

[1]Laboratorio de Microbiología Molecular, One Health Institute, Facultad de Ciencias de la Vida, Universidad Andres Bello, Santiago, Chile

[2]Laboratorio de Microbiología Aplicada y Extremófilos, Departamento de Ingeniería Química, Universidad Católica del Norte, Antofagasta, Chile

[3]Centro de Investigación Tecnológica del Agua y Sustentabilidad en el Desierto-CEIT-SAZA, Universidad Católica del Norte, Antofagasta, Chile

[4]Bio-computing and Genome Biology Laboratory, Center for Bioinformatics and Integrative Biology, Facultad de Ciencias de la Vida, Universidad Andres Bello, Santiago, Chile

[5]Instituto de Ciencias Biomédicas, Facultad de Medicina, Universidad de Chile, Santiago, Chile

[6]Instituto de Ciencias Biomédicas, Facultad de Medicina y Facultad de Ciencias de la Vida, Universidad Andres Bello, Santiago, Chile

[7]Laboratorio de Virología Celular y Molecular, Instituto de Ciencias Biomédicas, Facultad de Medicina, Universidad de Chile, Santiago, Chile

[8]Grupo de Resistencia Antimicrobiana en Bacterias Patógenas y Ambientales (GRABPA), Instituto de Biología, Pontificia Universidad Católica de Valparaíso, Valparaíso, Chile

## AUTHOR ORCIDs

Guillermo Valdivia http://orcid.org/0009-0002-1216-9439
Coral Pardo-Esté https://orcid.org/0000-0003-0958-3268
Gloria Arriagada https://orcid.org/0000-0002-9935-0451
Fernando Valiente-Echeverria https://orcid.org/0000-0001-9156-2516
Jorge H. Valdes http://orcid.org/0000-0002-9615-4973
Claudia P. Saavedra http://orcid.org/0000-0002-4248-8556

## FUNDING

| Funder | Grant(s) | Author(s) |
| --- | --- | --- |
| Agencia Nacional de Investigación y Desarrollo | ANILLO ATE220007 | Claudia P. Saavedra |
| Agencia Nacional de Investigación y Desarrollo | FONDECYT Regular 1250419 | Claudia P. Saavedra |

## AUTHOR CONTRIBUTIONS

Juan Castro-Severyn, Conceptualization, Data curation, Formal analysis, Investigation, Methodology, Software, Supervision, Validation, Visualization, Writing – original draft, Writing – review and editing | Nicolás Pacheco, Conceptualization, Data curation, Investigation, Methodology, Project administration, Resources, Validation, Writing – review and editing | Guillermo Valdivia, Data curation, Formal analysis, Methodology, Software, Visualization | Gabriel I. Krüger, Data curation, Formal analysis, Investigation, Methodology, Software, Supervision, Validation, Visualization, Writing – review and editing | Coral Pardo-Esté, Formal analysis, Investigation, Methodology, Visualization, Writing – original draft, Writing – review and editing | Francisco Remonsellez, Funding acquisition, Writing – review and editing | Aldo Gaggero, Funding acquisition, Writing – review and editing | Gloria Arriagada, Investigation, Writing – original draft, Writing – review and editing | Fernando Valiente-Echeverria, Funding acquisition, Writing – review and editing | Jorge Olivares-Pacheco, Conceptualization, Investigation, Writing – original draft, Writing – review and editing | Jorge H. Valdes, Conceptualization, Data curation, Formal analysis, Funding acquisition, Investigation, Supervision, Writing – original draft, Writing – review and editing | Claudia P. Saavedra, Conceptualization, Funding acquisition, Investigation, Project administration, Resources, Supervision, Writing – original draft, Writing – review and editing

## DATA AVAILABILITY

All the sequences analyzed in this research can be found deposited in DDBJ/ENA/GenBank under Bioproject PRJNA1222636.

## ETHICS APPROVAL

The study was approved by the Institutional Review Board (IRB) of Andres Bello University (folio 001b-2023), and written informed consent was obtained from the parents or legal guardians of all participants.

## ADDITIONAL FILES

The following material is available online.

### Supplemental Material

**Figure S1 (mSystems00467-25-s0001.pdf).** Boxplot comparing the relative abundances of the most abundant genera in oral (blue) and nasopharyngeal (orange) samples across three sampling times per group for each defined cluster.

### Open Peer Review

**PEER REVIEW HISTORY (review-history.pdf).** An accounting of the reviewer comments and feedback.

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
