## [Reviewer comments · mSystems]

Impact of Seasonal Variation on the Oral and Nasopharyngeal Microbiome in School-Aged Children: The School MicroBE initiative

Juan Castro-Severyn, Nicolás Pacheco, Guillermo Valdivia, Gabriel Krüger, Coral Pardo-Esté, Francisco Remonsellez, Aldo Gaggero, Gloria Arriagada, Fernando Valiente-Echeverria, Jorge Olivares Pacheco, Jorge Valdes, and Claudia Saavedra

Corresponding Author(s): Claudia Saavedra, Universidad Andres Bello

Review Timeline:

Submission Date:	March 31, 2025
Editorial Decision:	May 6, 2025
Revision Received:	June 6, 2025
Accepted:	June 30, 2025

Editor: Lindsay Kalan

Reviewer(s): The reviewers have opted to remain anonymous.

Transaction Report:

DOI: <https://doi.org/10.1128/msystems.00467-25>

Re: mSystems00467-25 (Impact of Seasonal Variation on the Oral and Nasopharyngeal Microbiome in School-Aged Children: The School MicroBE initiative)

Dear Dr. Claudia P. Saavedra:

Revision Guidelines

Sincerely,
Lindsay Kalan
Editor
mSystems

Reviewer #1 (Comments for the Author):

In this work, the authors aim to investigate changes in nasopharyngeal and oral microbiota among school-aged children in relation to change in seasonality. The authors utilized a longitudinal cohort of 119 children aged 4-13 years, collected paired nasopharyngeal and oral microbiomes samples for 16s rRNA gene sequencing, and analyzed community composition changes at each body site at three time points (autumn, winter, spring) over one year. They identified an increase in alpha diversity of

nasopharyngeal, but not oral samples, from autumn to spring. Using clustering and transition analyses, they describe specific associations among genera that change at each site with seasons. Overall, the paper provides insights on naso-oral microbiome changes in young children that may be associated with seasonal changes through a longitudinal study design. One key limitation of the study is lack of health status tracking, such as incidence of respiratory illnesses across seasons that impact and confound naso-oral microbiome structures. Likewise, use of 16s rRNA gene sequencing provides limited species-level insight on microbial community dynamics and, without concomitant bacterial/viral pathogen data, the study offers limited insight into the interactions of microbiota and pathogen carriage. There are limited sub-analyses by age cohorts, which could offer further insight on development of microbiome from early to late childhood.

- Major
- 1) Please clarify which "chronic respiratory conditions" were used as exclusion criteria in the study (Line 531).
 - 2) The authors did not appear to include positive or negative sequencing controls in the study, which would improve removal of contaminating sequences and provide greater validity of obtained sequencing data. If available, provide details of controls used.
 - 3) In the third section of Table 1, it would be informative to include median ages for the groups of samples collected in Autumn vs Winter vs Spring. Naso-oral microbiota remain dynamic throughout early to late childhood, thus if there is uneven sampling of ages across seasons, this may confound results.
 - 4) The finding of higher nasopharyngeal versus oral microbiome diversity is, in general, not consistent with prior investigations and warrants further discussion. In addition, at lines 263-265, the opposite conclusion is implied from sub-analysis of participants with samples collected across all three seasons. Lastly, at lines 375-77, the authors incorrectly reference the findings of a study which in fact identified oral microbiome diversity to be higher than nasal microbial diversity. Please address this incorrect citation and clarify discussion of presented findings.
 - 5) The differences in relative abundance of taxa are presented as changes in percentage over seasons, without accompanying statistical comparison. At least for key taxa, it would be informative to include statistical tests performed with p-values, even if non-significant, when describing these trends. There is one test included (line 225), though it is unclear what specific age group was analyzed.
 - 6) Lines 486-488: This statement is not supported by the presented data, as there was no significant difference in diversity of the oral microbiome with season.

- Minor
- 1) Repeat use of "in children" at line 72.
 - 2) At line 129 and Table 1, the number of participants included in the study differs (n=119) compared to the abstract (n=120).
 - 3) Line 321-22: Please include which transitions were significant in the main text and/or Figure 8.
 - 4) Line 411: As written, this implies that all *Proteus* species are recognized as human pathogens, which is not the case. Modify to state "members of this genus [...]" In addition, urinary tract infections are commonly caused by *Proteus* spp. yet are not included in lines 414-15.
 - 5) Line 428: Refers to species, when later discussion pertains only to genus-level data in the transition network analyses.
 - 6) The authors note that samples were "transported to the laboratory within two hours of collection" however it would also be helpful to include details regarding sample storage prior to and following DNA extraction.

Reviewer #2 (Comments for the Author):

Reviewer's comments: Impact of Seasonal Variation on the Oral and Nasopharyngeal Microbiome in School-Aged Children: The School MicroBE initiative

General:

The authors present a study on the impact of seasonal variation on the oral and nasopharyngeal microbiome in school-aged children through a longitudinal study during which oral and nasopharyngeal samples were taken throughout different seasons. Notably, this study adds to the understanding of microbial community state types in these body sites, presenting a baseline study on healthy children. Of particular value is the longitudinal aspect of the study, which allows investigation of the stability of the different community types. Documenting and understanding these dynamics in healthy children will enable better public health strategies and targeted monitoring and e.g. improve microbiome-targeting therapies.

In general, the appropriate analysis methods are used, and the paper is well written, focusing on the nasal and oral microbiomes, which are relatively understudied in comparison to e.g. the gut microbiome, yet very important for (public) health. The authors point out a number of limitations of the study (single school sample, only bacteria,..etc.), which are fair to take into account and could form the basis for follow up work.

In addition to the scientific side, it is important that this work engaged children in an inspiring scientific project.

The current manuscript could be improved by further validating the clustering and transition analyses, examining potential DNA extraction kit contamination, and providing more information on the methodology used in this study.

Major comments:

1. Clustering and transition analysis: Apart from the method applied in the current version of the manuscript, the authors should test other clustering methods and compare them to validate the clusters and strengthen their conclusion on the presence of different clusters (e.g. as proposed in <https://doi.org/10.1371/journal.pcbi.1002863> and more recent works).
2. Oral samples: The methods for oral sampling should be more detailed; was sampling done before/after meals? Did

participants rinse their mouth prior to sampling? Did the participants brush their teeth prior to sampling?

3. As the authors also point out, *Herbaspirillum* is typically a soil microbe and a known contaminant in DNA extraction kits. In addition, *Tardiphaga* also typically found in soil samples and *Phyllobacterium* is a known contaminant in DNA extraction kits. This is especially an issue in low biomass samples (e.g. nasal swabs). The authors should motivate how they controlled for the "kitome" presence in their samples.

Minor comments:

1. Bioinformatic analysis: Information on sequencing depth and extraction controls is missing and should be included.
2. A follow up of the bacterial density (or an approximation thereof) in the nasopharyngeal samples would have been interesting to observe shifts in absolute abundances as well relative abundances. Has this been done/attempted during the study, e.g. by cell counts, qPCR/ddPCR, ...?
3. It would be interesting for the authors to add whether any effects on the similarity of the nasopharyngeal and oral microbiomes observed in children from the same family and/or classroom.
4. The authors could explain why less samples were taken in autumn; was this due to acute respiratory illnesses?
5. Typically, there are different microbial communities present in the mouth. In this study, different sites were sampled with one swab, which could result in a greater microbial diversity. Could the authors motivate why?
6. Data analysis in general: for reproducibility, it would be good to provide the code used for the data analysis.
7. Clustering; it could be useful to compare the clusters found in this study to others (doi: 10.3389/fmicb.2017.02372, 10.1126/sciadv.1400216), given the many similarities.
8. Were additional data on the children's health, oral hygiene or dietary habits etc. collected in the scope of this study, and if yes, which associations were observed with these microbiomes (if any)?

Spelling, grammar, and other comments:

1. L72: "in children" is written twice in the sentence
2. L141: "build environment" should be "built environment"
3. L215: "Spring" should be "spring"
4. L268: "Proteus" should be in italic
5. L605: Conflict of interest: It seems like the authors forgot to remove the standard text.
6. L636: "children" should be "children's"
7. Figures: Genus names should be in italic.
8. Figures: Legibility: The sample names/individual numbers in several of the barplots are in very small font sizes, which could be increased for legibility.
9. Figures: Axis labels: Verify that all plots have axis labels, e.g. sample name/individual number.
10. Throughout the text: As there are international differences in the system of grades in schools, providing the reader with the ages of the group, rather than the grade, would make the manuscript easier to read.

Reviewer #3 (Comments for the Author):

There are a few suggestions that would improve the manuscript.

Major comments:

1. The data comes from Chile, which is underrepresented in the microbiome literature. The authors refer to seasons by universal terms 'spring, autumn, winter' throughout the manuscript, but the readership of this journal is global, and assumptions about climate and time of year are heavily governed by an individual's location. Consider including the actual month for each 'season' to clarify this for readers in both hemispheres. For example, instead of using 'Spring' consider using 'Spring (October)'. This is critical so that further references and interpretations of the text by others regarding seasonality/climate are accurate. This should be done on both text and figures anytime a season is mentioned.
2. Because seasons are terms that evoke other factors like climate and lifestyle choices, I strongly suggest adding a section to the main text that describes the contextual setting. By this I mean a description of what time of year does each season occur in Chile, what is the average temperature during each one, average humidity, precipitation, etc. Are student exposures more likely to be impacted by the outdoors at certain times vs. other times of year? Student diets? Is there a certain time of year that school students are more likely to catch various viral respiratory infections? The authors allude to aspects of this in the text on occasion, but it is important enough to warrant its own section. Microbiomes reflect environments, as the authors state at several points, and giving readers more context will not only make the results easier to interpret, but critically this will aid future readers and/or other investigators who wish to reference this work later. This will add significantly to the impact of the manuscript.

Minor comments:

Line 247 - 'gender' is used instead of 'sex' when 'sex' is used up until this point. Please address.

Line 259 - consider adding the sample size of participants who were evaluated at all three seasons to the sentence for clarity on the size of the subset that this section of results refers to.

Lines 267-280 - this results section would be considerably enhanced by an 'environmental context' section as I suggest above in major comment #2.

Line 404 - Please clarify where these results came from, reference 30 or 31?

Figure 1 - please include y-axes and add months to season labels. Also, annotations should be added to the top and bottom boxes to clarify what they are showing.

Figure 10 - the legend says that central nodes correspond to groups of samples and the arrows correspond to individual samples that transition from one group to another between seasons - however the arrows are emerging from the central nodes themselves. Please clarify if this is trying to indicate that simply 'an' individual moves from one group to another or if it is supposed to show movement of specific individuals.

Reviewer's comments: Impact of Seasonal Variation on the Oral and Nasopharyngeal Microbiome in School-Aged Children: The School MicroBE initiative

General:

The authors present a study on the impact of seasonal variation on the oral and nasopharyngeal microbiome in school-aged children through a longitudinal study during which oral and nasopharyngeal samples were taken throughout different seasons.

Notably, this study adds to the understanding of microbial community state types in these body sites, presenting a baseline study on healthy children. Of particular value is the longitudinal aspect of the study, which allows investigation of the stability of the different community types. Documenting and understanding these dynamics in healthy children will enable better public health strategies and targeted monitoring and e.g. improve microbiome-targeting therapies.

In general, the appropriate analysis methods are used, and the paper is well written, focusing on the nasal and oral microbiomes, which are relatively understudied in comparison to e.g. the gut microbiome, yet very important for (public) health. The authors point out a number of limitations of the study (single school sample, only bacteria,..etc.), which are fair to take into account and could form the basis for follow up work.

In addition to the scientific side, it is important that this work engaged children in an inspiring scientific project.

The current manuscript could be improved by further validating the clustering and transition analyses, examining potential DNA extraction kit contamination, and providing more information on the methodology used in this study.

Major comments:

1. Clustering and transition analysis: Apart from the method applied in the current version of the manuscript, the authors should test other clustering methods and compare them to validate the clusters and strengthen their conclusion on the presence of different clusters (e.g. as proposed in <https://doi.org/10.1371/journal.pcbi.1002863> and more recent works).
2. Oral samples: The methods for oral sampling should be more detailed; was sampling done before/after meals? Did participants rinse their mouth prior to sampling? Did the participants brush their teeth prior to sampling?
3. As the authors also point out, *Herbaspirillum* is typically a soil microbe and a known contaminant in DNA extraction kits. In addition, *Tardiphaga* also typically

found in soil samples and *Phyllobacterium* is a known contaminant in DNA extraction kits. This is especially an issue in low biomass samples (e.g. nasal swabs). The authors should motivate how they controlled for the “kitome” presence in their samples.

Minor comments:

1. Bioinformatic analysis: Information on sequencing depth and extraction controls is missing and should be included.
2. A follow up of the bacterial density (or an approximation thereof) in the nasopharyngeal samples would have been interesting to observe shifts in absolute abundances as well relative abundances. Has this been done/attempted during the study, e.g. by cell counts, qPCR/ddPCR, ...?
3. It would be interesting for the authors to add whether any effects on the similarity of the nasopharyngeal and oral microbiomes observed in children from the same family and/or classroom.
4. The authors could explain why less samples were taken in autumn; was this due to acute respiratory illnesses?
5. Typically, there are different microbial communities present in the mouth. In this study, different sites were sampled with one swab, which could result in a greater microbial diversity. Could the authors motivate why?
6. Data analysis in general: for reproducibility, it would be good to provide the code used for the data analysis.
7. Clustering; it could be useful to compare the clusters found in this study to others (doi: [10.3389/fmicb.2017.02372](https://doi.org/10.3389/fmicb.2017.02372), [10.1126/sciadv.1400216](https://doi.org/10.1126/sciadv.1400216)), given the many similarities.
8. Were additional data on the children’s health, oral hygiene or dietary habits etc. collected in the scope of this study, and if yes, which associations were observed with these microbiomes (if any)?

Spelling, grammar, and other comments:

1. **L72:** “in children” is written twice in the sentence
2. **L141:** “build environment” should be “built environment”
3. **L215:** “Spring” should be “spring”
4. **L268:** “Proteus” should be in italic
5. **L605:** Conflict of interest: It seems like the authors forgot to remove the standard text.
6. **L636:** “children” should be “children’s”
7. Figures: Genus names should be in italic.

8. Figures: Legibility: The sample names/individual numbers in several of the barplots are in very small font sizes, which could be increased for legibility.
9. Figures: Axis labels: Verify that all plots have axis labels, e.g. sample name/individual number.
10. Throughout the text: As there are international differences in the system of grades in schools, providing the reader with the ages of the group, rather than the grade, would make the manuscript easier to read.

We would like to thank you and the reviewers for the constructive comments, critiques, and suggestions, which have significantly improved our manuscript titled "*Impact of Seasonal Variation on the Oral and Nasopharyngeal Microbiome in School-Aged Children: The School MicroBE Initiative.*"

Enclosed, please find the revised version of the manuscript for consideration in *mSystems*.

We have addressed all reviewer comments and incorporated the additional information requested. A point-by-point response to each issue raised is included, along with a marked-up version of the manuscript highlighting all changes made in this resubmission (hereafter referred to as the Revised Manuscript, RM).

RESPONSE TO REVIEWERS

Reviewer #1 (Comments for the Author):

In this work, the authors aim to investigate changes in nasopharyngeal and oral microbiota among school-aged children in relation to change in seasonality. The authors utilized a longitudinal cohort of 119 children aged 4-13 years, collected paired nasopharyngeal and oral microbiomes samples for 16s rRNA gene sequencing, and analyzed community composition changes at each body site at three time points (autumn, winter, spring) over one year. They identified an increase in alpha diversity of nasopharyngeal, but not oral samples, from autumn to spring. Using clustering and transition analyses, they describe specific associations among genera that change at each site with seasons. Overall, the paper provides insights on naso-oral microbiome changes in young children that may be associated with seasonal changes through a longitudinal study design. One key limitation of the study is lack of health status tracking, such as incidence of respiratory illnesses across seasons that impact and confound naso-oral microbiome structures. Likewise, use of 16s rRNA gene sequencing provides limited species-level insight on microbial community dynamics and, without concomitant bacterial/viral pathogen data, the study offers limited insight into the interactions of microbiota and pathogen carriage. There are limited sub-analyses by age cohorts, which could offer further insight on development of microbiome from early to late childhood.

We agree with what the reviewer points out regarding limitations of the current study. This work represents the first stage of the initiative, being a metataxonomic analysis the first approach to assess microbial composition profiles in school children, an unprecedented initiative in Chile and other South American countries. This study is focused on the first year of sampling, which aims to generate the baseline for future "in-deep" studies. The of 16S sequencing limitations, regarding the inability to identify microorganisms at the level species, including the inability to collect gene function information, are

compensated by the characterization of a large cohort of students, making our work a cost-effective approach that can be scaled up to larger populations.

For the assessment of children's health status, we conducted a comprehensive questionnaire to gather robust metadata (such as vaccinations, antibiotic usage, medical history) for all participants. However, we need to consider that no information about ill children is included, which may hamper potential associations with disease outcomes. Nevertheless, we strongly believe that the data presented is necessary to develop more informative and disease prevention-oriented studies.

Major

1) Please clarify which "chronic respiratory conditions" were used as exclusion criteria in the study (Line 531).

Yes, this specific information regarding asthma and chronic obstructive pulmonary disease has been added to the RM (see lines 551-552).

2) The authors did not appear to include positive or negative sequencing controls in the study, which would improve removal of contaminating sequences and provide greater validity of obtained sequencing data. If available, provide details of controls used.

We carry out several contamination checkpoints throughout the experimental design. We performed DNA extractions from environmental samples taken during the children sampling campaigns, during DNA extractions in the lab, and also with nuclease-free water to test whether the extraction kit itself could be a source of contamination. There were no detectable DNA concentrations for all cases with these controls (quantified with the Qubit™ 1X dsDNA HS Assay Kit from Thermo). Nonetheless, using these samples (without detectable DNA) as a template, we performed conventional PCR with the 16S rRNA universal primers 27F and 1515R, and there was no amplification, which we considered a reason to continue with our analysis and trust our data.

3) In the third section of Table 1, it would be informative to include median ages for the groups of samples collected in Autumn vs Winter vs Spring. Naso-oral microbiota remain dynamic throughout early to late childhood, thus if there is uneven sampling of ages across seasons, this may confound results.

We acknowledge that the naso-oral microbiota is highly dynamic and further develops during childhood, thus, age distribution among groups sampled in the different seasons could be a source of bias and a confounding factor. To address this concern, we calculated detailed descriptive statistics, including

median age, for the children’s samples obtained in each season and for each anatomical site (nasopharyngeal and oral).

Sample Type	Season	Mean	S.D.	Min.	Max.	Median
Nasopharyngeal	Autumn	6.500	1.358	5	10	6
	Winter	9.472	1.980	5	13	10
	Spring	8.614	2.251	5	13	9
Oral	Autumn	6.294	1.425	4	10	6
	Winter	9.271	2.172	5	13	10
	Spring	8.714	2.291	5	13	9

Considering these observed differences in median ages between seasons, we tested with additional statistical test the impact of age and its possible interaction with season on microbial composition. Specifically, we performed an Analysis of Variance (ANOVA) considering age (both as a continuous and a discrete variable) and season as factors, as well as their interaction. Additionally, we calculated the effect size using Eta-squared (η^2) to quantify the proportion of variance explained by these factors in the microbial diversity analyses. The results of these analyses are presented and discussed below.

Nasopharyngeal				Oral			
Tax_Count		p-value	η^2	Tax_Count		PR(>F)	η^2
Age Continue	Season	2.12E-33	0.583	Age Continue	Season	2.62E-39	0.6775
	Age	0.1990	0.0096		Age	0.0057	0.0477
	Season:Age	0.6729	0.0046		Season:Age	0.4970	0.0089
Age Discrete	Season	6.47.E-34	0.5952	Age Discrete	Season	1.66E-39	0.6864
	Age	0.4450	0.0095		Age	0.0013	0.0826
	Season:Age	0.0515	0.0539		Season:Age	0.7219	0.0133
Shannon		p-value	η^2	Shannon		p-value	η^2
Age Continue	Season	0.012987	0.0493	Age Continue	Season	0.0107	0.0562
	Age	0.293199	0.0064		Age	0.8598	0.0002
	Season:Age	0.385368	0.011		Season:Age	0.3600	0.0129
Age Discrete	Season	0.009261	0.0539	Age Discrete	Season	0.0099	0.0582
	Age	0.020151	0.0452		Age	0.2498	0.0179
	Season:Age	0.019072	0.0669		Season:Age	0.2739	0.0326
Simpson		p-value	η^2	Simpson		p-value	η^2
Age Continue	Season	0.092809	0.0273	Age Continue	Season	0.0032	0.0707
	Age	0.355102	0.005		Age	0.1030	0.0168
	Season:Age	0.844458	0.002		Season:Age	0.4274	0.0108
Age Discrete	Season	0.077583	0.0298	Age Discrete	Season	0.0030	0.0726
	Age	0.037658	0.0381		Age	0.0609	0.0357
	Season:Age	0.031986	0.0602		Season:Age	0.5052	0.0212

The ANOVA results show that the p-values for the Age factor or for the Age:Season interaction are non-significant in most of the microbial diversity analyses. Even if some p-values for the age factor or age:season interaction were statistically significant, the corresponding Eta-squared values are consistently low. This indicates that although there are differences in median ages among seasonal samples, age itself (or its interaction with season) explains only a small proportion of the observed variability in the microbiota.

In other words, Age would have a low impact or relevance as an explanatory factor for changes in the microbiota compared to other factors, such as season (which does show larger eta-squared values compared to age). Hence, these results justify that, despite variations in the age of the groups sampled across seasons, the age variable is not a primary factor in the reported microbiota changes. This validates the initial seasonal comparisons, with season being a more relevant influencing factor than an intrinsic variable of the studied population. This information has been added to the RM (see Table 1 and lines 165-167).

4) The finding of higher nasopharyngeal versus oral microbiome diversity is, in general, not consistent with prior investigations and warrants further discussion. In addition, at lines 263-265, the opposite conclusion is implied from sub-analysis of participants with samples collected across all three seasons. Lastly, at lines 375-77, the authors incorrectly reference the findings of a study which in fact identified oral

microbiome diversity to be higher than nasal microbial diversity. Please address this incorrect citation and clarify discussion of presented findings.

- **Correct, in general terms we find greater diversity (although not always significant) in nasopharyngeal samples, but we must also take into consideration that our data set comes from a heterogeneous population (Figure 2) that might have more sources of variability. Nonetheless, there are recent reports showing the variability of both anatomical sites. This variation can be explained by environmental causes such as diet, age, and health conditions. Also, it has been shown that by comparing oral and nasal microbiomes of adults and children, the effect of body site is higher than the effect of age, although the nasal microbiota is more variable between adults and children than the oral microbiota, leading to the conclusion that the latter develops at a slower rate.**
 - Zelasko S, Swaney MH, Sandstrom S, Davenport TC, Seroogy CM, Gern JE, et al. Upper respiratory microbial communities of healthy populations are shaped by niche and age. *Microbiome*. 2024 Oct 18;12(1):206.
 - Lei, Y., Li, M., Zhang, H., Deng, Y., Dong, X., Chen, P., ... & Tao, R. (2025). Comparative analysis of the human microbiome from four different regions of China and machine learning-based geographical inference. *Mosphere*, 10(1), e00672-24.
 - Gan, W., Yang, F., Meng, J., Liu, F., Liu, S., & Xian, J. (2021). Comparing the nasal bacterial microbiome diversity of allergic rhinitis, chronic rhinosinusitis and control subjects. *European Archives of Oto-Rhino-Laryngology*, 278, 711-718.

Nonetheless, we have modified the statement to improve clarity on the RM (see lines 271-274).

- **This is true and that was exactly what we found, we are aware that those 11 children are representative of the whole data set, but those were the only ones sampled on the three seasons.**
- **Yes, thank you for noticing this mistake, the paper we intended to reference was Wang, J., Feng, J., Zhu, Y., Li, D., Wang, J., & Chi, W. (2022). Diversity and biogeography of human oral saliva microbial communities revealed by the earth microbiome project. *Frontiers in Microbiology*, 13, 931065. in which the authors describe the nasopharyngeal microbiome as more diverse compared to the oral one (represented by saliva samples), in addition to correcting the citation we modified the paragraph to improve clarity (see lines 388-390 and 725-727).**

5) The differences in relative abundance of taxa are presented as changes in percentage over seasons, without accompanying statistical comparison. At least for key taxa, it would be informative to include statistical tests performed with p-values, even if non-significant, when describing these trends. There is one test included (line 225), though it is unclear what specific age group was analyzed.

We use the Wilcoxon rank-sum test for all the comparisons regarding the relative abundance of the dominant genera between anatomical sites and seasons; this clarification has been added to the RM to improve understanding (see lines 233-234).

6) Lines 486-488: This statement is not supported by the presented data, as there was no significant difference in diversity of the oral microbiome with season.

Thanks for noticing this; it was a mistake. The statement was intended to refer to the nasopharyngeal microbiomes, where a significant increase in diversity was observed. This was corrected in the RM (see line 506).

Minor

1) Repeat use of "in children" at line 72.

Thanks for noticing this, the error was corrected in the RM (see line 77).

2) At line 129 and Table 1, the number of participants included in the study differs (n=119) compared to the abstract (n=120).

Thank you for noticing this, there were indeed 120 children, but later one had to be eliminated due to poor sequencing coverage. This was corrected in the RM abstract (see line 36).

3) Line 321-22: Please include which transitions were significant in the main text and/or Figure 8.

We assume that the reviewer refers to Figure 10, which corresponds to the transition networks in the manuscript. The requested information has been added to the RM as well as modification to Figure 10 (see Figure 10 and lines 327-334).

4) Line 411: As written, this implies that all *Proteus* species are recognized as human pathogens, which is not the case. Modify to state "members of this genus [...]" In addition, urinary tract infections are commonly caused by *Proteus* spp. yet are not included in lines 414-15.

This paragraph has been modified to improve clarity and to include the reviewer's suggestion (see RM lines 426-432).

5) Line 428: Refers to species, when later discussion pertains only to genus-level data in the transition network analyses.

This was corrected in the RM (see lines 444, 445, 449).

6) The authors note that samples were "transported to the laboratory within two hours of collection" however it would also be helpful to include details regarding sample storage prior to and following DNA extraction.

DNA extractions were performed immediately on the same sampling day, and the remaining sample was stored as backup at -80°C in the laboratory. This information was added to the RM for clarity (see lines 578 and 582-583).

Reviewer #2 (Comments for the Author):

General:

The authors present a study on the impact of seasonal variation on the oral and nasopharyngeal microbiome in school-aged children through a longitudinal study during which oral and nasopharyngeal samples were taken throughout different seasons. Notably, this study adds to the understanding of microbial community state types in these body sites, presenting a baseline study on healthy children. Of particular value is the longitudinal aspect of the study, which allows investigation of the stability of the different community types. Documenting and understanding these dynamics in healthy children will enable better public health strategies and targeted monitoring and e.g. improve microbiome-targeting therapies. In general, the appropriate analysis methods are used, and the paper is well written, focusing on the nasal and oral microbiomes, which are relatively understudied in comparison to e.g. the gut microbiome, yet very important for (public) health. The authors point out a number of limitations of the study (single school sample, only bacteria,..etc.), which are fair to take into account and could form the basis for follow up work. In addition to the scientific side, it is important that this work engaged children in an inspiring scientific project. The current manuscript could be improved by further validating the clustering and transition analyses, examining potential DNA extraction kit contamination, and providing more information on the methodology used in this study.

We thank the reviewer for his comments and suggestions, which have helped to substantially improve the manuscript. In the RM all changes could be found marked in yellow.

Major comments:

1. Clustering and transition analysis: Apart from the method applied in the current version of the manuscript, the authors should test other clustering methods and compare them to validate the clusters and strengthen their conclusion on the presence of different clusters (e.g. as proposed in <https://doi.org/10.1371/journal.pcbi.1002863> and more recent works).

We thank the reviewer for this suggestion regarding the exploration of alternative clustering methods to validate our findings. We recognize the importance of ensuring the robustness of the identified clusters and, consequently, we have conducted additional analyses to address this concern. Following the reviewer's recommendation, we have tested several clustering methods to compare the results with the hierarchical clustering approach presented in the manuscript. Specifically, we evaluated:

1. Hierarchical Clustering (k=4) using Principal Coordinates Analysis (PCoA) ordination with Hellinger transformation for relative abundance data (Figure 1A).
2. Hierarchical Clustering (n=4) using PCoA ordination and Bray-Curtis distance (Figure 1B).
3. Partitioning Around Medoids (PAM, k=4) with Bray-Curtis distance and PCoA ordination (Figure 1C).
4. Partitioning Around Medoids (PAM, k=4) with Jensen-Shannon distance and PCoA ordination (Figure 1D).

Our main interest focused on validating the existence and composition of the most prominent and dense clusters identified in our original analysis, particularly those enriched in key taxa such as *Streptococcus* (belonging to Cluster N-XIII for the nasopharyngeal microbiome and Cluster O-V for the oral microbiome, as described in the manuscript). In the attached figure, we have highlighted with circles the analogous clusters that consistently emerged across these different methods.

Figure 1. Validation of microbiome cluster structure using different clustering approaches.

Visually, it can be observed that, although the number of defined clusters and the dispersion of some points may vary slightly between methods (which is expected given the nature of each algorithm and distance metrics), the main and most densely populated clusters show remarkable consistency regarding the groups of samples that compose them.

To formally assess the similarity between these main clusters identified by the different methods, we performed statistical comparisons. Specifically, we compared the sample composition within the highlighted clusters obtained through the different approaches. For this, we applied a Permutational Multivariate Analysis of Variance (PERMANOVA) on the matrices of taxon abundances present in each sample identified in each cluster for each method. The results of these permutation tests for the comparisons between the main clusters yielded p-values > 0.9999. This indicates that there are no statistically significant differences in the overall microbial community structure of these main clusters when identified by the different clustering methods tested.

This congruence between different clustering algorithms and distance metrics for the most prominent clusters provides strong validation of their existence and supports the robustness of our initial findings based on hierarchical clustering. While each method may offer different perspectives on finer-scale sub-clustering or the definition of sparse clusters, the consistent identification of these main and well-defined clusters (particularly those dominated by *Streptococcus*) across these diverse approaches strengthens our conclusions about the presence of these distinct microbiome states.

2. Oral samples: The methods for oral sampling should be more detailed; was sampling done before/after meals? Did participants rinse their mouth prior to sampling? Did the participants brush their teeth prior to sampling?

The samples were taken just before the children attended their breakfast break. Children did not rinse their mouths previous to the sampling, apart from just brushing their teeth when they woke up at home. Information to make this clearer has been added to the RM (see lines 572-574).

3. As the authors also point out, *Herbaspirillum* is typically a soil microbe and a known contaminant in DNA extraction kits. In addition, *Tardiphaga* also typically found in soil samples and *Phyllobacterium* is a known contaminant in DNA extraction kits. This is especially an issue in low biomass samples (e.g. nasal swabs). The authors should motivate how they controlled for the "kitome" presence in their samples.

***Herbaspirillum* has been widely associated with soils, but its association with pathogenicity and infections is currently increasing, in healthy or immunocompromised people, especially in rural and peripheral areas such as the case of our studied school:**

- Josy Panikulam, E., Seenarain, A., Shrestha, H., Leung Wan Chin, J., Jahan, T., Yap, G., ... & Meher-Homji, Z. (2025). A case report of *Herbaspirillum* infection in rural Australia. *ASM Case Reports*, 1(3), e00091-24.
- Li, X., Bao, X., Qiao, G., Wang, L., Shi, C., Chen, S., ... & Wang, Z. (2022). First study of bacteremia caused by *Herbaspirillum huttiense* in China: a brief research report and literature review. *Frontiers in Cellular and Infection Microbiology*, 12, 882827.

Similarly, *Phyllobacterium* has been largely associated with plants and soils in samples from different types of environments:

- Park, Y., Ten, L. N., Maeng, S., Chang, Y., Jung, H. Y., & Kim, M. K. (2021). *Phyllobacterium pellucidum* sp. nov., isolated from soil. *Archives of Microbiology*, 203, 2647-2652.
- Bromfield, E. S., Cloutier, S., & Hynes, M. F. (2024). *Phyllobacterium meliloti* sp. nov. a novel non-symbiotic bacterium isolated from root nodules of *Melilotus albus* (white sweet clover) grown in Canada. *bioRxiv*, 2024-11.

However, there are reports that have associated *Phyllobacterium* with oral and nasopharyngeal microbiomes, particularly in children infected with seasonal viruses such as influenza A, so it would not be so rare to find this taxon in our samples:

- Zhou, Q., Xie, G., Liu, Y., Wang, H., Yang, Y., Shen, K., ... & Zheng, Y. (2020). Different nasopharynx and oropharynx microbiota imbalance in children with *Mycoplasma pneumoniae* or influenza virus infection. *Microbial pathogenesis*, 144, 104189.
- Wen, Z., Xie, G., Zhou, Q., Qiu, C., Li, J., Hu, Q., ... & Wen, F. (2018). Distinct nasopharyngeal and oropharyngeal microbiota of children with influenza A virus compared with healthy children. *BioMed research international*, 2018(1), 6362716.

While *Tardiphaga* is still mostly unknown, there are few reports where it has been identified in soil samples, cold environments and associated with plants:

- Guro, P. V., Sazanova, A. L., Kuznetsova, I. G., Tikhomirova, N. Y., Belimov, A. A., Yakubov, V. V., & Safronova, V. I. (2023). Genetic diversity of root nodule endophyte strains isolated from the legumes *Astragalus umbellatus* and *A. inopinatus*, growing on the Kamchatka Peninsula (Russian Federation). *Russian Journal of Plant Physiology*, 70(8), 185.
- Bao, Z., Wang, C., Cao, J., Zhang, T., Guo, Y., Sato, Y., ... & Ohta, H. (2024). *Tardiphaga alba* sp. nov., a heavy-metal-tolerant bacterium isolated from garden soil. *International Journal of Systematic and Evolutionary Microbiology*, 74(1), 006238.

Although we could not find reports associating this genus with the human microbiome, we highlight that the studied school is located in a peripheral area of Santiago, and the school facilities include yards, playgrounds, and gardens, where children spend their breaks possibly gathering environmental microbes as part of their regular daily activities, making possible the presence of this taxonomic groups in the samples.

In addition, we were very meticulous throughout the work, checking environmental controls during sample collection and DNA extraction protocols. We perform DNA extraction from the swabs and tubes we used to collect the samples, from the laboratory environmental samples taken while extracting the DNA, and also from a blank to test the kit. In all cases, no detectable DNA was found using Qubit's high-sensitivity kit. Finally, we used all those controls without detectable DNA to amplify the 16S rRNA gene with the universal primers 27F and 1525R, and we did not observe any amplicon when running the agarose gel. After all this, we sent our samples to be sequenced with the greatest possible certainty of not having contamination.

Minor comments:

1. Bioinformatic analysis: Information on sequencing depth and extraction controls is missing and should be included.

We thank the reviewer for the suggestion, this information has been added to the RM (see lines 150 and 581-583).

2. A follow up of the bacterial density (or an approximation thereof) in the nasopharyngeal samples would have been interesting to observe shifts in absolute abundances as well relative abundances. Has this been done/attempted during the study, e.g. by cell counts, qPCR/ddPCR, ...?

We didn't consider doing this in this initial phase, as there were many samples and limited resources. In any case, in the next phase of the project, we propose using other approaches, such as respiratory virus panels for all samples and metagenomic analysis of selected samples.

3. It would be interesting for the authors to add whether any effects on the similarity of the nasopharyngeal and oral microbiomes observed in children from the same family and/or classroom.

Indeed, this is a very interesting approach. Unfortunately, we don't have samples from the same family circle, and the number of children we were able to enroll per classroom is low. However, we performed analyses comparing classrooms and the results showed no tendencies, patterns nor big differences, so we discarded them. It's possible that in the second stage of the project, with a larger sample size, we can address this with greater confidence.

4. The authors could explain why less samples were taken in autumn; was this due to acute respiratory illnesses?

Yes, that's our guess. Around that time, children missed more school, and that's what happened. On the day we sampled our enrolled children, many were absent. This may be related to seasonal respiratory virus peaks. We'll address this in our next manuscript, where we'll try to correlate the microbiome's composition patterns to the results of the respiratory viral panel assay we performed on each child.

5. Typically, there are different microbial communities present in the mouth. In this study, different sites were sampled with one swab, which could result in a greater microbial diversity. Could the authors motivate why?

Yes, precisely because we were trying to capture the greatest possible diversity of the entire oral cavity, we sampled the tongue, gums, palate and the inner part

of the cheeks, so we call this sample oral. Unlike other studies where only the inner part of the cheeks is sampled and this is referred to as a buccal sample.

6. Clustering; it could be useful to compare the clusters found in this study to others (doi: 10.3389/fmicb.2017.02372, 10.1126/sciadv.1400216), given the many similarities.

Following the reviewer's suggestion, this comparison was included in the discussion section (see RM lines 489-493).

7. Were additional data on the children's health, oral hygiene or dietary habits etc. collected in the scope of this study, and if yes, which associations were observed with these microbiomes (if any)?

Yes, we conduct a comprehensive survey and questionnaire when we enroll children for the study, trying to obtain as much information as possible regarding habits, diet, health history, and socioeconomic status. Unfortunately, many children and their parents did not answer many of the questions due to lack of knowledge, poor memory, or an unwillingness to provide so much information. This resulted in a data set that we cannot fully trust. However, we used it for many analyses, but we did not find any kind of pattern, for example, with the vaccination program or whether there were smokers in the home.

Spelling, grammar, and other comments:

1. L72: "in children" is written twice in the sentence

This mistake was fixed in the RM (see line 77).

2. L141: "build environment" should be "built environment"

This mistake was fixed in the RM (see line 145).

3. L215: "Spring" should be "spring"

This mistake was fixed in the RM (see line 218).

4. L268: "Proteus" should be in italic

This mistake was fixed in the RM (see line 277).

5. L605: Conflict of interest: It seems like the authors forgot to remove the standard text.

This mistake was fixed in the RM (see lines 641-643).

6. L636: "children" should be "children's"

The mistake has been corrected in the revised version of the RM (see line 669).

7. Figures: Legibility: The sample names/individual numbers in several of the barplots are in very small font sizes, which could be increased for legibility.

It is not quite possible to enlarge the names much more without altering the graphics, which would no longer fit on one page. In any case, the final version will have high-quality, high-resolution figures that can be zoomed in as much as necessary in the electronic version without losing resolution.

8. Figures: Axis labels: Verify that all plots have axis labels, e.g. sample name/individual number.

Thank you very much, this has been meticulously reviewed throughout the entire manuscript.

9. Throughout the text: As there are international differences in the system of grades in schools, providing the reader with the ages of the group, rather than the grade, would make the manuscript easier to read.

This was considered throughout the RM (see lines 136, 164, 224-226, 232).

Reviewer #3 (Comments for the Author):

There are a few suggestions that would improve the manuscript.

Major comments:

1. The data comes from Chile, which is underrepresented in the microbiome literature. The authors refer to seasons by universal terms 'spring, autumn, winter' throughout the manuscript, but the readership of this journal is global, and assumptions about climate and time of year are heavily governed by an individual's location. Consider including the actual month for each 'season' to clarify this for readers in both hemispheres. For example, instead of using 'Spring' consider using 'Spring (October)'. This is critical so that further references and interpretations of the text by others regarding seasonality/climate are accurate. This should be done on both text and figures anytime a season is mentioned.

We want to establish and describe seasonal changes, and this topic was a major discussion among the work authors. However, in the methodology section this is indicated: the exact month to which each sample corresponds according to the season (see RM lines 558-559).

2. Because seasons are terms that evoke other factors like climate and lifestyle choices, I strongly suggest adding a section to the main text that describes the contextual setting. By this I mean a description of what time of year does each season occur in Chile, what is the average temperature during each one, average humidity, precipitation, etc. Are student exposures more likely to be impacted by the outdoors at certain times vs. other times of year? Student diets? Is there a certain time of year that school students are more likely to catch various viral respiratory infections? The authors allude to aspects of this in the text on occasion, but it is important enough to warrant its own section. Microbiomes reflect environments, as the authors state at several points, and giving readers more context will not only make the results easier to interpret, but critically this will aid future readers and/or other investigators who wish to reference this work later. This will add significantly to the impact of the manuscript. **We fully agree with the reviewer and have incorporated the suggestion. For this purpose, climatological information has been added to the RM (see lines 560-567).**

Minor comments:

Line 247 - 'gender' is used instead of 'sex' when 'sex' is used up until this point. Please address.

This mistake was fixed in the RM (see line Table 1).

Line 259 - consider adding the sample size of participants who were evaluated at all three seasons to the sentence for clarity on the size of the subset that this section of results refers to.

This information has been added (see RM lines 267-268).

Lines 267-280 - this results section would be considerably enhanced by an 'environmental context' section as I suggest above in major comment #2.

This has been considered and added to the discussion in the RM (see lines 421-423).

Line 404 - Please clarify where these results came from, reference 30 or 31?

In this paragraph we refer to both studies, in reference 31 the authors analyzed samples from children with adenoids and otitis media, while in reference 30 the authors address the nasopharyngeal microbiota in children with lower respiratory tract infections (see RM lines 413-414).

Figure 1 - please include y-axes and add months to season labels. Also, annotations should be added to the top and bottom boxes to clarify what they are showing.

The figure has been adapted incorporating the reviewer's suggestions (see Figure 1 in the RM).

Figure 10 - the legend says that central nodes correspond to groups of samples and the arrows correspond to individual samples that transition from one group to another between seasons - however the arrows are emerging from the central nodes themselves. Please clarify if this is trying to indicate that simply 'an' individual moves from one group to another or if it is supposed to show movement of specific individuals. **The figure legend has been adapted and information has been added to clarify what the reviewer pointed out (see Figure 10 and legend in the RM lines 354-357).**

Re: mSystems00467-25R1 (Impact of Seasonal Variation on the Oral and Nasopharyngeal Microbiome in School-Aged Children: The School MicroBE initiative)

Dear Dr. Claudia P. Saavedra:

Your manuscript has been accepted, and I am forwarding it to the ASM production staff for publication. Your paper will first be checked to make sure all elements meet the technical requirements. ASM staff will contact you if anything needs to be revised before copyediting and production can begin. Otherwise, you will be notified when your proofs are ready to be viewed.

Cover Image Submissions: If you would like to submit a potential Cover Image, please email a file and a short legend to mSystems@asmusa.org. Please note that we can only consider images that (i) the authors created or own and (ii) have not been previously published. By submitting, you agree that the image can be used under the same terms as the published article. Image File requirements: TIF/EPS, 7.5 inches wide by 8.25 inches tall (at least 2,250 pixels wide by 2,475 pixels tall), minimum 300 dpi resolution (600 dpi preferred), RGB, and no figure elements, e.g., arrows or panel labels. The legend should be a short description of the image, 1-2 sentences recommended. Please download and use this interactive template in Adobe to ensure that your proposed cover image meets our size requirements (<https://journals.asm.org/pb-assets/pdf-text-excel-files/ASM-Interactive-Sizing-Cover-Template-1715689791.pdf>).

Sincerely,
Lindsay Kalan
Editor
mSystems

Reviewer #1 (Comments for the Author):

All comments have been addressed and the manuscript/figures revised, as appropriate.

Reviewer #2 (Comments for the Author):

I would like to thank the authors for integrating the reviewer's comments in their manuscript and look forward to the follow-up papers of this project

Reviewer #3 (Comments for the Author):

Thank you for your attention to the reviewer comments!

All of my critiques have been addressed satisfactorily, and I feel that the authors were responsive to all comments given. The manuscript is much improved, and I think it is suitable for publication.